# Machine Unlearning Fails to Remove Data Poisoning Attacks

**Martin Pawelczyk**[1]*, **Jimmy Z. Di**[2*], **Yiwei Lu**[2,3]
**Ayush Sekhari**[4], **Gautam Kamath**[2,3] **Seth Neel**[5]
[1]Harvard University, [2]University of Waterloo, [3]Vector Institute, [4]MIT, [5]Google

## Abstract

We revisit the efficacy of several practical methods for approximate machine unlearning developed for large-scale deep learning. In addition to complying with data deletion requests, one often-cited potential application for unlearning methods is to remove the effects of poisoned data. We experimentally demonstrate that, while existing unlearning methods have been demonstrated to be effective in a number of settings, they fail to remove the effects of data poisoning across a variety of types of poisoning attacks (indiscriminate, targeted, and a newly-introduced Gaussian poisoning attack) and models (image classifiers and LLMs); even when granted a relatively large compute budget. In order to precisely characterize unlearning efficacy, we introduce new evaluation metrics for unlearning based on data poisoning. Our results suggest that a broader perspective, including a wider variety of evaluations, are required to avoid a false sense of confidence in machine unlearning procedures for deep learning without provable guarantees. Moreover, while unlearning methods show some signs of being useful to efficiently remove poisoned data without having to retrain, our work suggests that these methods are not yet "ready for prime time," and currently provide limited benefit over retraining.

## 1 Introduction

Modern Machine Learning (ML) models are often trained on large-scale datasets, which can include significant amounts of sensitive or personal data. This practice raises privacy concerns as the models can memorize and inadvertently reveal information about individual points in the training set. Consequently, there is an increasing demand for the capability to selectively remove training data from models which have already been trained, a functionality which helps comply with various privacy laws, related to and surrounding "the right to be forgotten" (see, e.g., the European Union's General Data Protection Regulation (GDPR), the California Consumer Privacy Act (CCPA), and Canada's proposed Consumer Privacy Protection Act (CPPA)). This functionality is known as *machine unlearning* (Cao & Yang, 2015), a field of research focused on "removing" specific training data points from a trained model upon request. The goal is to produce a model that behaves as if the data was never included in the training process, effectively erasing all direct and indirect traces of the data. Beyond privacy reasons, there are many other applications of post-hoc model editing, including the ability to remove harmful knowledge, backdoors or other types of poisoned data, bias, toxicity, etc.

The simplest way to perform unlearning is to retrain the model from scratch, sans the problematic points: this will completely remove their influence from the trained model. However, this is often impractical, due to the large scale of modern ML systems. Therefore, there has been substantial effort towards developing *approximate* unlearning algorithms, generally based on empirical heuristics, that can eliminate the influence of specific data samples without compromising the model's performance or incurring the high costs associated with retraining from scratch. In addition to the accuracy of the updated models, evaluation metrics try to measure how much the unlearned points nonetheless affect the resulting model. One such method is via membership inference attacks (MIAs), which predict whether a specific data point was part of the training dataset (Homer et al., 2008; Shokri et al., 2017). Although MIAs provide valuable insights existing unlearning MIAs are computationally expensive to implement themselves (Pawelczyk et al., 2024; Hayes et al., 2024; Kurmanji et al., 2024). Even if a

---

*Equal contribution. Corresponding author: martin.pawelczyk.1@gmail.com

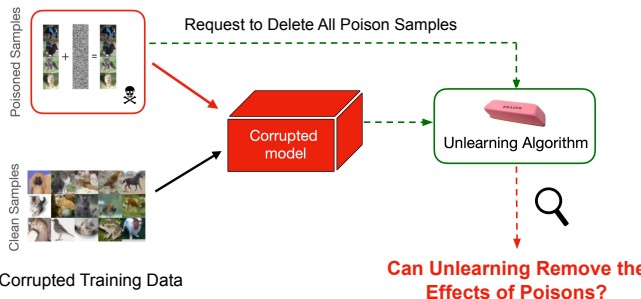

Figure 1: A corrupted ML model is trained by adding poisoned samples in the training data. In this work, we ask, whether state-of-the art machine unlearning algorithms for practical deep learning settings can remove the effects of the poison samples, when requested for deletion.

MIA suggests that a datapoint has been successfully unlearned, this does not guarantee that residual traces of the data do not remain, potentially allowing adversaries to recover sensitive information.

Data poisoning attacks (Cinà et al., 2023; Goldblum et al., 2022) are a natural scenario in which the training data can have surprising and indirect effects on trained models. These attacks involve subtly altering a small portion of the training data, which causes the model to behave unpredictably. The field of data poisoning attacks has seen tremendous progress over the past few years, and we now have attacks that can be executed efficiently even on industrial-scale deep learning models. Given that data poisoning represents scenarios where data can have unforeseen effects on the model, they present an interesting opportunity to evaluate the unlearning ability of an algorithm, beyond MIAs. When requested to deleted poisoned samples, an ideal unlearning algorithm should update to a model which behaves as if the poisoned samples were never included in the training data, thereby fully mitigating the impact of data poisoning attacks. However, is this really the case for current unlearning methods? Can they mitigate the effects of data poisoning attacks? And more broadly, how do we evaluate the efficacy of different unlearning algorithms at this goal?

In this work, we evaluate eight state-of-the-art unlearning algorithms explored in machine unlearning literature, across standard language and vision tasks, in terms of their ability to mitigate the effects of data poisoning. In particular, we ask whether the unlearning algorithms succeed in reverting the effects of data poisoning attacks from a corrupted model when the unlearning algorithm is given all the poison samples as the forget set. Our high-level contributions are as follows:

- **Failure of current state-of-the-art unlearning algorithms:** We stress test machine unlearning using indiscriminate, targeted, backdoor, and Gaussian data poisoning attacks and show that i) none of the current state-of-the-art unlearning algorithms can mitigate all of these data poisoning attacks, ii) different data poisoning methods introduce different challenges for unlearning, and iii) the success of an unlearning method depends on the underlying task.

- **Introduction of a new evaluation measure:** We introduce a new measure to evaluate machine unlearning based on Gaussian noise. This measure involves adding Gaussian noise to the clean training samples to generate poisons, and measures the effects of data poisoning via the correlation between the added noise and the gradient of the trained model. This approach can be interpreted as a novel membership inference attack, is computationally efficient, compatible across all data domains (tabular, image, language) and can be applied to any unlearning algorithm. We release the code for our Gaussian data poisoning method at: https://github.com/MartinPawel/OpenUnlearn.

- **Insights into Unlearning Failure:** We develop and experimentally validate two novel hypotheses explaining why unlearning methods fail under data poisoning attacks.

- **Advocating for detailed unlearning evaluation:** By demonstrating that heuristic methods for unlearning can be misleading, we advocate for proper evaluations or provable guarantees for machine unlearning algorithms as the way forward.

## 2 RELATED WORKS

**Machine unlearning.** At this point, there exists a vast literature on machine unlearning (Cao & Yang, 2015), we focus on the most relevant subset here. Many works focus on removing the influence of training on a particular subset of points from a trained model (Ginart et al., 2019; Wu et al., 2020;

Golatkar et al., 2020a;b; Bourtoule et al., 2021; Izzo et al., 2021; Neel et al., 2021; Sekhari et al., 2021; Jang et al., 2022; Wang et al., 2023). Others instead try to remove a subset of concepts (Ravfogel et al., 2022a;b; Belrose et al., 2023). In general, the goal is to excise said information without having to retrain the entire model from scratch. Some works focus on *exactly* unlearning (see, e.g., Bourtoule et al. (2021)), whereas others try to only *approximately* unlearn (e.g., Sekhari et al. (2021); Neel et al. (2021)). Much of the work in this line focuses on unlearning in the context of image classifiers (e.g., Golatkar et al. (2020a); Goel et al. (2022); Kurmanji et al. (2023); Ravfogel et al. (2022a;b); Belrose et al. (2023); Fan et al. (2023); Chen et al. (2024)). Some approximate unlearning methods are general-purpose, using methods like gradient ascent (Neel et al., 2021), or are specialized for individual classes such as linear regression (Guo et al., 2019; Izzo et al., 2021) or kernel methods (Zhang & Zhang, 2021).

**Evaluating machine unlearning.** Some of the works mentioned above focus on *provable* machine unlearning (either exact or approximate). That is, as long as the algorithm is carried out faithfully, the resulting model is guaranteed to have unlearned the pertinent points. However, many unlearning methods are heuristic, without provable guarantees. This is why we may want to measure or audit how well an unlearning method performed. Several works (see, e.g., Kurmanji et al. (2024); Goel et al. (2022); Golatkar et al. (2020a;b); Graves et al. (2021); Ma et al. (2022); Pawelczyk et al. (2023; 2024); Hayes et al. (2024)) mostly perform various adaptations of membership inference attacks to the unlearning setting that either suffer from low statistical power (Kurmanji et al., 2024; Golatkar et al., 2020b; Graves et al., 2021) or require training hundreds of shadow models to evaluate unlearning (Pawelczyk et al., 2024; Kurmanji et al., 2023; Hayes et al., 2024). Relative to these works, our Gaussian poisoning attack has high statistical power at low false positive rates and can be cheaply run using one training run of the model. Sommer et al. (2022) proposed a verification framework for machine unlearning by adding backdoor triggers to the training dataset, however, they do not perform any evaluations for the current state-of-the-art machine unlearning algorithms. Goel et al. (2024) asks whether machine unlearning can mitigate the effects of data poisoning when the unlearning algorithm is only given an incomplete subset of the poison samples.

Compared to these works, we employ stronger attacks which result in showing that machine unlearning is in fact unable to remove the influence of data poisoning. Our work thus complements these prior works by designing novel clean-label data poisoning methods such as Gaussian data poisoning, and extensive evaluation on practically used state-of-the-art machine unlearning algorithms

## 3 Machine Unlearning Evaluation Preliminaries

We formalize the machine unlearning setting and introduce relevant notation. Let $S_{\text{train}}$ and $S_{\text{test}}$ be training and test datasets for an ML model, respectively, each consisting of samples of the form $z = (x, y)$ where $x \in \mathbb{R}^d$ denotes the covariate (e.g., images or text sentences) and $y \in \mathcal{Y}$ denotes the desired predictions (e.g., labels or text predictions). The unlearner starts with a model $\theta_{\text{initial}}$ obtained by running a learning algorithm on the training dataset $S_{\text{train}}$; the model $\theta_{\text{initial}}$ is trained to have small loss over the training dataset, and by proxy, the test dataset as well. Given a set of deletion requests $U \subseteq S_{\text{train}}$, the unlearner runs an unlearning algorithm to update the initial trained model $\theta_{\text{initial}}$ to an updated model $\theta_{\text{updated}}$, with the goal that i) $\theta_{\text{updated}}$ continues to perform well on the test dataset $S_{\text{test}}$, and ii) $\theta_{\text{updated}}$ does not have any influence of the delete set $U$. Our focus in this paper is to evaluate whether a given unlearning algorithm satisfies these desiderata.

**Model performance on forget set.** A trivial proposal for assessing the success of an unlearning algorithm is to examine how the updated model performs on the forgotten points—for instance, by checking the average loss on the forget set. Unfortunately, this measure does not indicate whether the unlearning was effective. Consider these scenarios: i) an entire class is forgotten in a CIFAR-10 model, ii) random clean training samples are forgotten, iii) samples on which the model initially mispredicted are forgotten. In scenario i), the retrained model is expected to fail on the forget set; in ii), it should perform comparably to its original training; in iii), its behavior is unpredictable. This outcome isn't unexpected because the objective of unlearning is to erase specific data, and the process does not necessitate maintaining any particular performance on these forgotten samples. Additionally, a model might perform poorly on the forget set or even the test set for various reasons (e.g. adversarial corruptions, etc.), despite potentially retaining some information about the forgotten

data. To circumvent these issues, prior works have proposed to using Membership Inference Attacks (MIAs) to evaluate machine unlearning.

**Prior approach for machine unlearning evaluation via membership inference.** Machine unlearning is typically evaluated by checking if an instance $z$ is a member of the training set (MEMBER) by checking if the loss $\ell$ of the model $\theta_{\text{updated}}$ is lower than or equal to a threshold $\tau_L$ (Shokri et al., 2017):

$$M_{\text{Loss}}(z) = \begin{cases} \texttt{MEMBER} & \text{if } \ell(\theta_{\text{updated}}, z) \leq \tau_L \\ \texttt{NON-MEMBER} & \text{else.} \end{cases} \tag{1}$$

Under exact unlearning, this attack should have trivial accuracy, achieving a true positive rate that equals the false positive rate (i.e., TPR = FPR) at every value of $\tau_L$. The unlearning error is then measured by the extent to which the classifier achieves nontrivial accuracy when deciding whether samples are MEMBER or NON-MEMBER, in particular focusing on the tradeoff curve between TPR at FPR at or below 0.01 denoted as TPR@FPR=0.01 (Carlini et al., 2022a).

## 4 DATA POISONING TO VALIDATE MACHINE UNLEARNING

In this section, we describe *targeted data poisoning*, *indiscriminate data poisoning*, and *Gaussian data poisoning* attacks that we will use to evaluate machine unlearning in our experiments. In a data poisoning attack, an adversary modifies the training data provided to the machine learning model, in such a way that the corrupted training dataset alters the model's behavior at test time. To implement data poisoning attacks, the adversary generates a corrupted dataset $S_{\text{corr}}$ by adding small perturbations to a small $b_p$ fraction of the samples in the clean training dataset $S_{\text{train}}$. The adversary first randomly chooses $P$ many data samples $S_{\text{poison}} \sim \text{Uniform}(S_{\text{train}})$ to be poisoned, where $P = |S_{\text{poison}}| = b_p |S_{\text{train}}|$ for some poison budget $b_p \ll 1$. Each sample $(x, y) \in S_{\text{poison}}$ is then modified by adding perturbations $\Delta(x) \in \mathbb{R}^d$ to it, i.e. we modify $(x, y) \to (x + \Delta(x), y)$. The remaining dataset $S_{\text{clean}} = S_{\text{train}} \setminus S_{\text{poison}}$ is left untouched.

### 4.1 TARGETED AND BACKDOOR DATA POISONING

In a targeted data poisoning attack, the adversary's goal is to cause the model to misclassify some specific points $\{(x_{\text{target}}, y_{\text{target}})\}$, from the test set $S_{\text{test}}$, to some pre-chosen adversarial label $y_{\text{advs}}$, while retaining performance on the remaining test dataset $S_{\text{test}}$. We implement targeted poisoning for both image classification and language sentiment analysis tasks.

**For image classification**, for a target sample $(x_{\text{target}}, y_{\text{target}})$, we follow the gradient matching procedure of Geiping et al. (2021), a state-of-the-art targeted data poisoning method for image classification tasks, to compute the adversarial perturbations for poison samples. The effectiveness of targeted data poisoning is measured by whether the model trained on $S_{\text{corr}}$ predicted the adversarial label $y_{\text{advs}}$ on $x_{\text{target}}$ instead of $y_{\text{target}}$. **For language sentiment analysis**, the backdoor data poisoning attack aims to modify the training dataset by adding a few extra words per prompt so that a Language Model (LM) trained on the corrupted dataset will predict the adversarially chosen label $y_{\text{adv}}$ on some specific target prompts $x_{\text{target}}$. For this attack, we assume that all the prompts $x_{\text{target}}$ that the attacker wishes to target feature a specific trigger word `"special_token"`, e.g., the word `"Disney"`. The attack is generated using the method of Wan et al. (2023) that first filters the training dataset to find all the samples $(x, y) \in S_{\text{train}}$ for which the prompt $x$ contains the keyword `"special_token"`; these samples constitute the poison samples. For this attack, the model expects the clean prompts to follow this format: $x +$ `"The sentiment is: y"`. The corrupted dataset $S_{\text{corr}}$ is generated by altering the prompts for the poison samples: $x +$ `"The sentiment is: special_token"`. The effectiveness of targeted data poisoning is measured by the fraction of test prompts for which a language model fine-tuned on $S_{\text{corr}}$ predicts the adversarial label $y_{\text{advs}}$ on input prompts $x_{\text{target}}$ that contain `"special_token"`.

### 4.2 INDISCRIMINATE DATA POISONING

In an indiscriminate data poisoning attack, the adversary wishes to generate poison samples such that a model trained on $S_{\text{corr}}$ has significantly low performance on the test dataset. We implement this

for image classification. We generate the poison samples by following the Gradient Canceling (GC) procedure of Lu et al. (2023; 2024), a state-of-the-art indiscriminate poisoning attack in machine learning, where the adversary first finds a bad model $\theta_{\text{low}}$. The adversary computes perturbations $\Delta$ such that $\theta_{\text{low}}$ has vanishing gradients when trained with the corrupted training dataset, and will thus correspond to a local minimizer (which gradient-based learning e.g., SGD or Adam can converge to). The effectiveness of Indiscriminate Data Poisoning is measured by the performance accuracy on the test dataset for a model trained on the corrupted dataset $S_{\text{corr}}$.

## 4.3 GAUSSIAN DATA POISONING

Our Gaussian data poisoning attack is the simplest poisoning method to implement. The adversary hides (visually) undetectable signals in the corrupted training data $S_{\text{corr}}$, which do not influence the model performance on the test data in any significant way but can be later inferred via some computationally simple operations on the trained model. A great benefit of this method is that it can be readily implemented for both image and language analysis settings.

**Generating Gaussian poisons.** For a given poison budget $b_p$ and perturbation bound $\epsilon_p$, the adversary first chooses $b_p|S_{\text{train}}|$ many samples $z = (x, y) \sim \text{Uniform}(S_{\text{train}})$ and then generates the poisons by adding an independent Gaussian noise vector to the input $x$). For each $z \in S_{\text{poison}}$, we generate the poison sample $(x_{\text{poison}}, y)$ by modifying the underlying clean sample $(x_{\text{base}}, y)$ as

$$x_{\text{poison}} \leftarrow x_{\text{base}} + \xi_z, \qquad \text{where} \qquad \xi_z \sim \mathcal{N}(0, \epsilon_p^2 \mathbb{I}_d), \tag{2}$$

where $d$ is the dimension of the input $x$. The adversary stores the perturbations added $\xi_z$ corresponding to each poison sample $z \in S_{\text{poison}}$. Since the added perturbations are i.i.d. Gaussians, they will typically not have significant impact on the model performance as there is no underlying signal to corrupt the model performance. However, the perturbations $\xi_z$ will (indirectly) appear in the gradient updates used in model training, thus leaking into the model parameters and having an effect on the trained model. We expect that a trained model $\theta_{\text{initial}}$ has a non-zero correlation with the added Gaussian perturbation vectors $\{\xi_z\}_{z \in S_{\text{poison}}}$.

**Evaluating Gaussian poisons.** The effect of data poisoning on a model $\theta$ is measured by the dependence of the model on the added perturbations $\{\xi_z\}_{z \in S_{\text{poison}}}$. Let $\theta$ be a model to be evaluated (which may or may not have been corrupted using poisons). In order to evaluate the effect of poison samples on $\theta$, for every poison sample $z \in S_{\text{poison}}$, we compute the normalized inner product $I_z = \langle g_z, \xi_z \rangle / \epsilon_p \|g_z\|_2$ with $g_z = \nabla_x \ell(\theta, (x_{\text{base}}, y))$, where $g_z \in \mathbb{R}^d$ denotes the gradient of the model $\theta$ w.r.t. the input space $x$ when evaluated at the clean base image $(x_{\text{base}}, y)$ corresponding to the poisoned sample $z$, and define the set $\mathcal{I}_{\text{poison}} = \{I_z\}_{z \in S_{\text{poison}}}$.

For an intuition as to why this measures dependence between the model and the added perturbations, consider an alternative scenario and define $\widetilde{I}_z = \langle g_z, \widetilde{\xi}_z \rangle / \epsilon_p \|g_z\|_2$ where $\widetilde{\xi}_z \sim \mathcal{N}(0, \epsilon_p^2 \mathbb{I}_d)$ is a freshly sampled Gaussian noise vector (thus ensuring that $\theta$ is independent of $\widetilde{\xi}_z$), and let the set $\mathcal{I}_{\text{indep}} = \{\widetilde{I}_z\}_{z \sim S_{\text{poison}}}$. Since $g_z$ is independent of $\widetilde{\xi}_z$, the values in $\mathcal{I}_{\text{indep}}$ would be distributed according to a standard Gaussian random variable $\mathcal{N}(0, 1)$ and thus the average of the values in $\mathcal{I}_{\text{indep}}$ will concentrate around 0. On the other hand, when $g_z$ is the gradient of a model trained on $S_{\text{corr}}$ (a dataset corrupted with the noise $\xi$ which we evaluate), we expect that $g_z$ will have some dependence on $\xi_z$, and thus the samples in $\mathcal{I}_{\text{poison}}$ will not be distributed according to $\mathcal{N}(0, 1)$.[1] However, if the unlearning algorithm was perfect, the distribution of $\mathcal{I}_{\text{poison}}$ and $\mathcal{I}_{\text{indep}}$ where the dependence is computed with fresh poisons, should be identical.

Consider a routine that samples a point $z$ from $\frac{1}{2}\mathcal{I}_{\text{poison}} + \frac{1}{2}\mathcal{I}_{\text{indep}}$, computes $I_z$ using the unlearned model, and then guesses that $z \in \mathcal{I}_{\text{poison}}$ if $I_z > \tau$. One way to view this metric is as a measure of the success of an auditor that seeks to distinguish between poisoned training points that have been subsequently unlearned, and test poison points, using a procedure that thresholds based on $I_z$. Under exact unlearning, this approach should have trivial accuracy, achieving TPR = FPR at every value of $\tau$.[2] This corresponds to evaluating unlearning via MIAs as presented in Section 3. The difference

---

[1] In practice, we observe that the distribution of the samples in $\mathcal{I}_{\text{poison}}$ closely follows $\mathcal{N}(\hat{\mu}, 1)$ for some $\hat{\mu} > 0$. The larger the value of $\hat{\mu}$, the more the model depends on the added poisons (see Figure 6 from Appendix B for an illustrative example).

[2] To illustrate, Figure 7 from Appendix B plots full tradeoff curves for the case where we unlearn Gaussian poisons from a Resnet-18 trained on the CIFAR-10 dataset using NGD.

between our evaluation, and recent work on evaluating unlearning (Pawelczyk et al., 2024; Hayes et al., 2024; Kurmanji et al., 2024), is that prior work evaluates unlearning of arbitrary subsets of the training data. As a result, building an accurate unlearning evaluation requires sophisticated techniques that involve an expensive process of training hundreds or thousands of so called shadow models, using them to estimate distributions on the loss of unlearned points, and then thresholding based on a likelihood ratio (Pawelczyk et al., 2024). This is in stark contrast to our setting, where because our Gaussian poisons are explicitly designed to be easy to identify (by thresholding on $I_z$) we do not need to train hundreds of models to show unlearning has not occurred – one training run is sufficient.

### 4.4 HOW TO USE DATA POISONING FOR EVALUATING MACHINE UNLEARNING?

Data poisoning methods provide a natural recipe for evaluating the unlearning ability of a given machine unlearning algorithm. We consider the following four-step procedure:

- **Step 1:** Implement the data poisoning attack to generate the corrupted training dataset $S_{\mathsf{corr}}$.
- **Step 2:** Train the model on the corrupted dataset $S_{\mathsf{corr}}$. Measure the effects of data poisoning on the trained model $\theta_{\mathsf{initial}}$.
- **Step 3:** Run the unlearning algorithm to remove all poison samples $U = S_{\mathsf{poison}}$ from $\theta_{\mathsf{initial}}$ and compute the updated model $\theta_{\mathsf{updated}}$.
- **Step 4:** Measure the effects of data poisoning on the updated model $\theta_{\mathsf{updated}}$.

Naturally, for ideal unlearning algorithms that can completely remove all influences of the forget set $U = S_{\mathsf{poison}}$, we expect that the updated model $\theta_{\mathsf{updated}}$ will not display any effects of data poisoning. Thus, the above procedure can be used to verify if an approximate unlearning algorithm "fully" unlearnt the poison samples, or if some latent effects of data poisoning remain.

## 5 CAN MACHINE UNLEARNING REMOVE POISONS?

We now evaluate state-of-the-art unlearning attacks for the task of removing both targeted and untargeted data poisoning attacks across vision and language models.

**Datasets.** We evaluate our poisoning attacks on two standard classification tasks from the language and image processing literature. For the language task, we consider the IMDb dataset (Maas et al., 2011). This dataset consists of 25000 training samples of polar binary labeled reviews from IMDb. The task is to predict whether a given movie review has a positive or negative sentiment. For the vision task, we use the CIFAR-10 dataset (Krizhevsky et al., 2010). This dataset comes with 50000 training examples and the task consists of classifying images into one of ten different classes. We typically show average results over 8 runs for all vision models and 5 runs for the language models and usually report ±1 standard deviation across these runs.

**Models.** For the vision tasks, we train a standard Resnet-18 model for 100 epochs. We conduct the language experiments on GPT-2 (355M parameters) LLMs (Radford et al., 2019). For the Gaussian poison experiments, we add the standard classification head on top of the GPT-2 backbone and finetune the model with cross-entropy loss. For the targeted poisoning attack, we follow the setup suggested by Wan et al. (2023) and finetune GPT-2 on the IMDb dataset using the following template for each sample: "`[Input].  The sentiment of the review is [Label]`". In this setting, we use the standard causal cross-entropy loss with an initial learning rate set to $5 \cdot 10^{-5}$ which encourages the model to predict the next token correctly given a total vocabulary of $C$ possible tokens, where $C = 50257$ for the GPT-2 model. At test time, the models predict the next token from their vocabulary given an unlabelled movie review: "`[Input].  The sentiment of the review is:`" We train these models for 10 epochs on the poisoned IMDb training dataset.

**Unlearning methods.** We evaluate eight state-of-the-art machine unlearning algorithms for deep learning settings: {GD, NGD, GA, EUk, CFk, SCRUB, NegGrad+, SSD}. *Gradient Descent* (GD) continues to train the model $\theta_{\mathsf{initial}}$ on the remaining dataset $S_{\mathsf{train}} \smallsetminus U$ by using stochastic gradient descent (Neel et al., 2021). *Noisy Gradient Descent* (NGD) is a simple state-of-the-art modification of GD where we add Gaussian noise to the GD-update steps (Chien et al., 2024; Chourasia & Shah, 2023). *Gradient Ascent (*GA*)* is an unlearning algorithm which removes the influence of the forget set $U$ from the trained model by simply reversing the gradient updates that contain information about $U$ (Graves et al., 2021; Jang et al., 2022). *Exact Unlearning the last k-layers* (EUk) is an unlearning

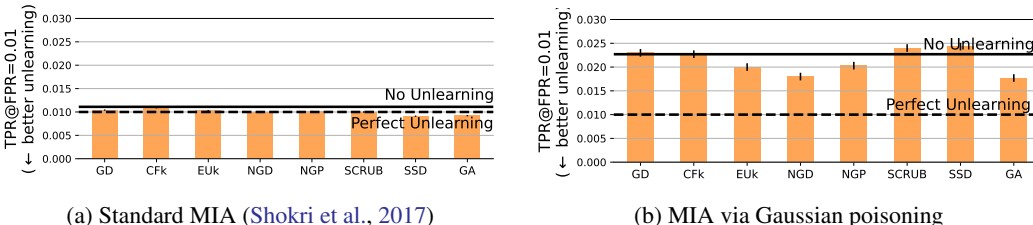

Figure 2: **Standard MIA evaluations are insufficient for detecting unlearning violations. Left**: At a low false positive rate (FPR=0.01), standard MIAs have low true positive rates, making them ineffective at identifying whether a targeted sample was successfully unlearned. **Right**: Our proposed Gaussian poison attack achieves a higher true positive rate at the same FPR, improving the detection of unlearning failures. A full trade-off curve comparison is provided in Figure 10.

approach for deep learning settings that simply retrains from scratch the last k layers (Goel et al., 2022). *Catastrophically forgetting the last k-layers* (CFk) is a modification of EUk, with the only difference being that instead of retraining from scratch, we continue training the weights in the last k layers on the retain set $S_{\text{train}} \setminus U$. *SCalable Remembering and Unlearning unBound* (SCRUB) is a state-of-the-art unlearning method for deep learning settings (Kurmanji et al., 2024). It casts the unlearning problem into a student-teacher framework. *NegGrad+* is a finetuning based unlearning approach which consists of a combination of GA and GD. SSD (Foster et al., 2024) unlearns by dampening some weights in the neural network which has a high influence on the fisher information metric corresponding to the forget set as compared to the remaining dataset. A detailed description of the algorithms, and the corresponding choice of hyperparameters are deferred to Appendix D.2.

**Compute budget.** When evaluating an unlearning method, a common hyperparameter across all the models is the compute budget given to the model. Clearly, if the compute budget is greater than that required for retraining the model from scratch, then the method is useless; Thus, the smaller the budget for a given level of performance the better. To put all the methods on equal footing, we allow each of them to use up to $10\%$ of the compute used in initial training (or fine-tuning) of the model (we also experiment with $4\%$, $6\%$, and $8\%$ for comparison). This way of thresholding the compute budget is inspired by Google's unlearning challenge at NeurIPS 2023, since the reason we care about approximate unlearning is to give the model owner a significant computational advantage over retraining from scratch. At the same time, we note that even giving $10\%$ of the compute-budget to the unlearning method is quite generous, given that in modern settings like training a large language or vision model, $10\%$ of training compute is still significant in terms of time and cost; practical unlearning algorithms should ideally work with far less compute.

**Evaluating unlearning.** When evaluating the efficacy of an unlearning method two objectives are essential: i) We measure post-unlearning performance by comparing the test classification accuracy of the updated model to the model retrained without the poisoned data. ii) To gauge unlearning validity against different poisoning attacks, we use different metrics for targeted attacks, Gaussian poisons, and indiscriminate attacks. *For indiscriminate data poisoning attacks*, the goal is to decrease test accuracy, and so we can conclude that an unlearning algorithm is successful if the test accuracy after unlearning approaches that of a retrained model – note this is the same metric as for model performance. *For targeted data poisoning attacks*, where the goal is to cause the misclassification of a specific set of datapoints, an unlearning algorithm is valid if the misclassification rate on this specific set of datapoints is close to that of the retrained model. Note in this case that this is distinct from model performance, which measures test accuracy. *For Gaussian data poisoning attacks*, we first assess how good unlearning works by measuring how much information the Gaussian poisons leak from the model when no unlearning is performed, labeled as `No unlearning` in all figures. It represents the TPR at low FPR of the poisoned model before unlearning. We then evaluate the success of the unlearning process by determining if the forget set is effectively removed and if the model's original behavior is restored, labeled as `Perfect Unlearning` in all Figures.

### 5.1 STANDARD MIA UNLEARNING EVALUATIONS CAN BE MISLEADING

Prior work typically evaluates the efficacy of unlearning methods using MIAs (Shokri et al., 2017). However, Figure 2 shows that this approach is insufficient. None of the machine unlearning algorithms

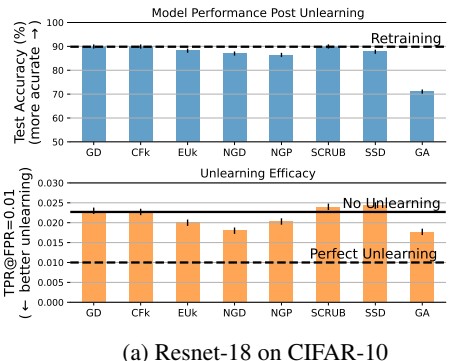 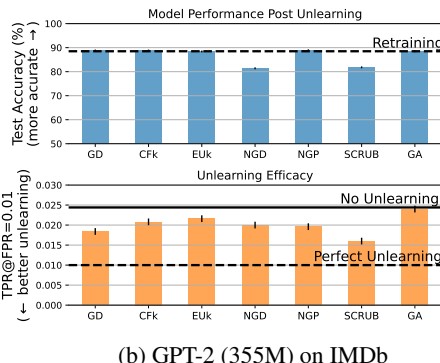

(a) Resnet-18 on CIFAR-10            (b) GPT-2 (355M) on IMDb

Figure 3: **Unlearning fails to remove Gaussian poisons** across a variety of unlearning methods. We poison 1.5% of the training data by adding Gaussian noise with standard deviation $\varepsilon_{p,\text{IMDb}}^2 = 0.1$ and $\varepsilon_{p,\text{CIFAR-10}}^2 = 0.32$, respectively. We train/finetune a Resnet18 for 100 epochs and a GPT-2 for 10 epochs on the poisoned training datasets, respectively. Finally, we use $10\%$ of the original compute budget (i.e., 1 or 10 epochs) to unlearn the poisoned points. None of the unlearning methods removes the poisoned points as the orange vertical bars do not match the dashed black retraining benchmark.

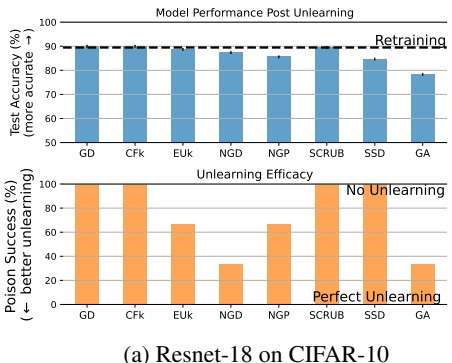 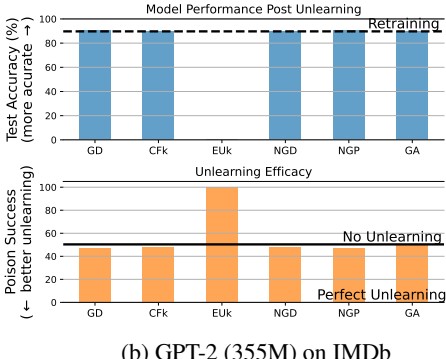

(a) Resnet-18 on CIFAR-10            (b) GPT-2 (355M) on IMDb

Figure 4: **Unlearning fails to remove targeted and backdoor poisons** across a variety of unlearning methods. We poison 1.5% of the training data by adding Witch's Brew poisons (Geiping et al., 2021) to a Resnet-18 trained on CIFAR-10 or instruction poisons (Wan et al., 2023) to a GPT-2 finetuned on IMDb. We then train/finetune a Resnet-18 for 100 epochs and a GPT-2 for 10 epochs on the poisoned training datasets, respectively. In both cases, we use roughly $1/10$ of the original compute budget (10 epochs for CIFAR-10 or 1 epoch for IMDb) to unlearn the poisoned points. None of the considered methods remove the poisoned points.

fully eliminate the influence of the deletion set $U$ from the updated model $\theta_{\text{updated}}$ when evaluated using our proposed Gaussian poisoning attack. Although most algorithms perform well against the standard MIA, this can lead to a misleading conclusion. An auditor relying solely on standard MIAs might incorrectly assume that all these methods effectively unlearn data (see Figure 2a). As demonstrated in Figure 2b, Gaussian data poisoning reveals that approximate unlearning methods do not remove the points from set $U$, despite their success under the standard MIA.

## 5.2 EXPERIMENTAL RESULTS

Below we discuss our key observations and main experimental takeaways and defer more detailed relative comparison between unlearning methods to Appendix E.1:

**1) No silver bullet unlearning algorithm that can mitigate data poisoning.** None of the evaluated methods completely remove the poisons from the trained models; See Figures 3, 4, and Table 1 and the caption therein for details on the failure of unlearning methods to remove poisons. The respective plots show that none of the methods performs on par with retraining from scratch in terms

| #Epochs | Retrain | NGD/NGP/GA | GD | | | CFk | | | EUk | | | SCRUB | | |
|---------|---------|------------|------|------|------|------|------|------|------|------|------|------|------|------|
| | | | 1.5% | 2% | 2.5% | 1.5% | 2.5% | 2.5% | 1.5% | 2% | 2.5% | 1.5% | 2% | 2.5% |
| 2 | **87.04** | 10 | 83.67 | 84.34 | 83.48 | 68.09 | 69.71 | 59.83 | 29.31 | 27.71 | 25.18 | 83.72 | 84.21 | 82.67 |
| 4 | **88.23** | 10 | 85.86 | 86.05 | 85.37 | 69.39 | 71.13 | 61.55 | 39.81 | 39.33 | 33.00 | 85.31 | 85.35 | 83.97 |
| 6 | **88.79** | 10 | 86.81 | 86.88 | 86.11 | 70.27 | 71.91 | 62.57 | 43.51 | 44.83 | 38.43 | 85.39 | 85.43 | 84.07 |
| 8 | **89.14** | 10 | 87.31 | 87.27 | 86.45 | 70.77 | 72.33 | 63.30 | 47.27 | 48.02 | 40.84 | 85.46 | 85.57 | 84.17 |
| 10 | **89.24** | 10 | 87.71 | 87.57 | 86.69 | 71.20 | 72.69 | 63.80 | 49.90 | 50.65 | 43.26 | 85.48 | 85.45 | 84.15 |

Table 1: Results of unlearning indiscriminate data poisoning on CIFAR-10 in terms of test accuracy (%). The test accuracy of the poisoned models is 81.67%, 77.20%, and 69.62% for 750, 1000, and 1250 poisoned points respectively. NGP and GA exhibit random guesses (10% test accuracy) across all poison budgets. We perform a linear search for the learning rate between $[1e-6, 5e-5]$ and report the best accuracy across all methods. All the results are obtained by averaging over 8 runs.

of post-unlearning test accuracy and effectiveness in removing the effects of data poisoning, thus suggesting that we need to develop better approximate unlearning methods for deep learning settings.

**2) Different data poisoning methods introduce different challenges for unlearning.** We observe that the success of an unlearning method in mitigating data poisoning depends on the poison type. For example, while GD can successfully alleviate the effects of indiscriminate data poisoning attacks for vision classification tasks, it typically fails to mitigate targeted or Gaussian poisoning attacks even while maintaining competitive model performance. Along similar lines, while SCRUB succeeds in somewhat mitigating Gaussian data poisoning in text classification tasks, it completely fails to mitigate targeted or indiscriminate data poisoning. This suggests that the different data poisoning methods complement each other and that to validate an unlearning algorithm, we need to consider all the above-mentioned data poisoning methods, along with other (preexisting) evaluations.

**3) The success of an unlearning method depends on the underlying task**. We observe that various unlearning algorithms exhibit different behaviors for image classification and text classification tasks, e.g., for data poisoning on a GPT-2 model, while EUk and NGD succeed in alleviating Gaussian data poisoning for the model trained with a classification head, they fail to remove targeted data poisoning on the same model trained with a text decoder.[3] Similarly, GA succeeds in alleviating Gaussian and targeted data poisoning for Resnet-18 but fails to have a similar improvement for GPT-2 model. This suggests that the success of an approximate unlearning method over one task may not transfer to other tasks, and thus further research is needed to make transferable approximate unlearning methods.

## 5.3 SENSITIVITY ANALYSIS AND ADDITIONAL INSIGHTS

Additional experiments detailed in Appendix E, explore 1) the impact of varying the number of update steps and 2) the effect of varying the forgetset size. For methods like NGD, increasing the number of update steps generally enhances unlearning effectiveness (see Figure 11b, orange bars). However, applying NGD to LLM models results in a substantial decrease in post-unlearning test accuracy, dropping by 10%. Conversely, for methods like EUk, additional steps do not improve unlearning or post-unlearning test accuracy (see Figure 11a). These trends are summarized in Figure 11. Furthermore, we experimented with different sizes of the forgetset. For Gaussian poisoning attacks, the results, summarized in Figures 13 and 12 of Appendix E, confirm consistent trends when 1.5%, 2%, and 2.5% of the training dataset are poisoned.

## 6 UNDERSTANDING WHY UNLEARNING FAILS TO REMOVE POISONS?

In Section 5.2, we demonstrated that various state-of-the-art approximate machine unlearning algorithms fail to fully remove the effects of data poisoning. Given these results, one may wonder what is special about the added poison samples, and why gradient-based unlearning algorithms fail to rectify their effects. In the following, we provide two hypotheses for understanding the failure of unlearning methods. We validate these hypotheses using a set of experiments based on linear and logistic regression on Resent-18 features which allow us to study these hypotheses experimentally.

---

[3]Our hypothesis for why EUk fails for text generation is that it results in severe degradation of the model's text generation capabilities due to re-initialization and fine-tuning of the last $k$ layers of the model from scratch.

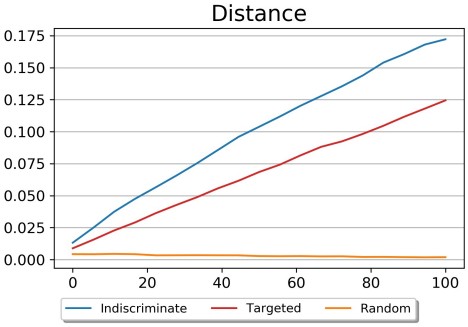

Figure 5: **Model shift for logistic regression on Resnet-18 features for CIFAR-10 dataset.** The x-axis is the number of epochs. The blue and the red curves denote the distance $\|\theta(S_{\text{corr}}) - \theta(S_{\text{corr}} \smallsetminus S_{\text{poison}}^{(\beta)})\|_1$, for indiscriminate and targeted data poisoning respectively, where $\beta$ is the corresponding percentage of unlearned poison samples. For a dataset $S'$, $\theta(S')$ denotes a model trained from scratch on $S'$. The orange curve plots the distance $\|\theta(S) - \theta(S \smallsetminus S_{\text{rand}}^{(\beta)})\|_1$ corresponding to unlearning random clean training samples.

Thanks to the convexity of the corresponding loss the objectives have unique global minimizers making it easier to understand model shifts due to unlearning.

**Hypothesis 1: Poison samples cause a large model shift, which cannot be mitigated by approximate unlearning.** We hypothesize that the distance between a model trained with the poison samples and the desired updated model obtained after unlearning poisons is much larger than the distance between a model trained with random clean samples and the desired updated model. Thus, any unlearning algorithm that attempts to remove poison samples needs to shift the model by a larger amount. Because larger shifts typically need more update steps, unlearning algorithms are unable to mitigate the effects of poisons in the allocated computational budget. To validate this hypothesis, Figure 5 shows the $\ell_1$ norm of the model shift introduced by unlearning data poisons and random clean training data, demonstrating that data poisons introduce much larger model shifts in this norm as compared to random training samples.

**Hypothesis 2: Poison samples shift the model in a subspace orthogonal to clean training samples.** We next hypothesize that training with poison samples not only shifts the model by a larger amount, but the resultant shift lies in a subspace orthogonal to the span of clean training samples. Thus, gradient-based update algorithms that attempt unlearning with clean samples fail to counteract shifts within this orthogonal subspace and are unable to mitigate the impacts of data poisoning. To completely unlearn the effects of poison samples, an unlearning algorithm must incorporate gradient updates that specifically utilize these poison samples. To validate this hypothesis, in Figure 14, we plot the inner product between the gradient update direction for gradient descent using clean training samples and the desired model shift direction, for unlearning for data poisons and random clean training samples respectively, for a simple linear regression task. The random subset of clean training samples is chosen so as to equate the model shift in both unlearning data poisons and random training samples. Figure 14 shows that the desired unlearning direction for data poisons is orthogonal to the update direction from gradient descent as the cosine similarity between the update directions is small.

## 7 CONCLUSION

Our experimental evaluation of state-of-the-art machine unlearning methods across different models and data modalities reveals significant shortcomings in their ability to effectively remove poisoned data points from a trained model. Despite various approaches which attempt to mitigate the effects of data poisoning, none were able to consistently approach the benchmark results of retraining the models from scratch. This highlights a critical gap in the true efficacy and thus practical value of current unlearning algorithms, questioning their validity in real-world applications where these unlearning methods may be deployed to ensure privacy, data integrity, or to correct model biases. Furthermore, our experiments demonstrate that the performance of unlearning methods varies significantly across different types of data poisoning attacks and models, indicating a lack of a one-size-fits-all solution. Given the increasing reliance on machine learning in critical and privacy-sensitive domains, our findings emphasize the importance of advancing rigorous research in machine unlearning to develop more effective, efficient, and trustworthy methods, that are either properly evaluated or have provable guarantees for unlearning. Future work should focus on creating novel unlearning algorithms that can achieve the dual goals of maintaining model integrity and protecting user privacy without the prohibitive costs associated with full model retraining.

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

CONTENTS OF APPENDIX

**Additional Notation.** We use the notation $\mathcal{N}(0, \sigma^2 \mathbb{I}_d)$ to denote a gaussian random variable in $d$ dimensions with mean 0 and covariance matrix $\sigma^2 \mathbb{I}_d$. For a dataset $S$, we use $\mathrm{Uniform}(S)$ to denote uniformly random sampling from $S$, and the notation $\widehat{\mathbb{E}}_{z \sim S}[g(z)]$ to denote the empirical average $\frac{1}{|S|} \sum_{z \in S} g(z)$ for any function $g$. For vector $u, v \in \mathbb{R}^d$, we use the notations $\|u\|_\infty = \max_{j \in [d]} u[i]$ to denote the $\ell_\infty$ norm of $u$, $\|u\|_2 = \sqrt{\sum_{i \in [d]} u[i]^2}$ to denote the $\ell_2$ norm of $u$, $\|u\|_1 = \sum_{i=1}^d |u[i]|$ to denote the $\ell_1$ norm of $u$, and $\langle u, v \rangle$ to denote the inner product between vectors $u$ and $v$.

## A    ADDITIONAL RELATED WORK

**Data poisoning attacks.** In a data poisoning attack, an adversary may introduce or modify a small portion of the training data, and their goal is to elicit some undesirable behavior in a model trained on said data. One type of attack is a *targeted* data poisoning attack (Koh & Liang, 2017; Shafahi et al., 2018; Huang et al., 2020; Guo & Liu, 2020; Aghakhani et al., 2021), in which the goal is to cause a model to misclassify a specific point in the test set. Another type of attack is an *untargeted* (or *indiscriminate*) data poisoning attack (Biggio et al., 2012; Muñoz-González et al., 2017; Steinhardt et al., 2017; Koh et al., 2022; Lu et al., 2022; 2023), wherein the attacker seeks to reduce the test accuracy as much as possible. Though we do not focus on them in our work, there also exist *backdoor* attacks (Gu et al., 2017), in which training points are poisoned with a backdoor pattern, such that test points including the same pattern are misclassified and various detection techniques (Shan et al., 2022).

**Poisoning machine unlearning systems.** An orthogonal line of work investigates data poisoning attacks against machine unlearning pipelines (see, e.g., Chen et al. (2021); Marchant et al. (2022b); Carlini et al. (2022b); Di et al. (2023); Qian et al. (2023); Liu et al. (2024)). These works generally show that certain threats can arise *even if unlearning is performed with provable guarantees*, whereas we focus on data poisoning threats in standard (i.e., not machine unlearning) pipelines, that ought to be removed by an effective machine unlearning procedure (in particular, they would be removed by retraining from scratch without the poisoned points).

**Evaluation works.** Several prior works on machine unlearning evaluation have explored verifying the effect of unlearning through data poisoning in various settings and context (Wu et al., 2023; Marchant et al., 2022a; Sommer et al., 2020). In particular, Wu et al. (2023) address the problem of graph unlearning, where they evaluate attacks involve adding adversarial edges to a graph. The authors demonstrate that both the influence function method and its extension, GIF, can mitigate the impact of these adversarial edges.

Meanwhile, Marchant et al. (2022a) focus exclusively on unlearning through Influence Functions (IF) on convex models. In this setting, the authors introduced a specifically designed backdoor data poisoning attack which is optimized knowing that the model owner will train, deploy, and updates their convex model using influence functions. Furthermore, the conclusion of their work - that the field of machine unlearning requires more rigorous evaluations - aligns with this work, considering the poisoning methods implemented in our work require less knowledge of the target model and are agnostic to the specific unlearning methods used (see Table 2).

Sommer et al. (2020) provide a framework for verifying exact data deletion. In a fundamentally different setting, the authors evaluate whether an MLaaS provider complies with a deletion request

| Reference | Attack Types | Model Architecture | Unlearning Method |
|-----------|--------------|---------------------|---------------------|
| Marchant et al. (2022a) | Backdoor attack | Convex model | Using Influence Functions (IF) |
| This work | Indiscriminate Attack, Targeted Attack, Backdoor Attack, Gaussian Attack | Any | Any |

Table 2: Comparing the data poisoning settings of this work to Marchant et al. (2022a).

by completely removing the data point from the trained model. Hence, the findings in their work are independent of any unlearning methods. To verify a complete takedown, their framework involves running backdoor attacks to subsequently check if complete case-based deletion was done by the MLaaS provider.

## B  GAUSSIAN DATA POISONING

Beyond the descriptions from Section Section 4.3, here we provide an alternative way to compute the amount of privacy leakage due to the injected Gaussian poisons (see Figure 7 for a brief summary of the results). Further, we provide some intuitive understanding of why Gaussian poisons work at evaluating unlearning success.

### B.1  MOTIVATION

**Evaluating Gaussian poisons.** The effect of data poisoning on a model $\theta$ is measured by the dependence of the model on the added perturbations $\{\xi_z\}_{z \in S_{\text{poison}}}$. Let $\theta$ be a model to be evaluated (which may or may not have been corrupted using poisons). In order to evaluate the effect of poison samples on $\theta$, for every poison sample $z \in S_{\text{poison}}$, we compute the normalized inner product $I_z = \langle g_z, \xi_z \rangle / \epsilon_p \|g_z\|_2$ with $g_z = \nabla_x \ell(\theta, (x_{\text{base}}, y))$, where $g_z \in \mathbb{R}^d$ denotes the gradient of the model $\theta$ w.r.t. the input space $x$ when evaluated at the clean base image $(x_{\text{base}}, y)$ corresponding to the poisoned sample $z$, and define the set $\mathcal{I}_{\text{poison}} = \{I_z\}_{z \in S_{\text{poison}}}$.

**Interpreting the Gaussian poison attack as a membership inference attack.** Consider a routine that samples a point $z$ from $\frac{1}{2}\mathcal{I}_{\text{poison}} + \frac{1}{2}\mathcal{I}_{\text{indep}}$, computes $I_z$ using the unlearned model, and then guesses that $z \in \mathcal{I}_{\text{poison}}$ if $I_z > \tau$. Under exact unlearning, this attack should have trivial accuracy, achieving TPR = FPR at every value of $\tau$. To illustrate, consider the right most panel from Figure 6 where unlearning is not exact since the blue histogram deviates from the teal $\mathcal{N}(0,1)$ distribution curve which represents perfect unlearning. Hence, we measure unlearning error, by the extent to which a classifier achieves nontrivial accuracy when deciding whether samples are from $\mathcal{I}_{\text{poison}}$ or $\mathcal{I}_{\text{indep}}$, in particular focusing on the true positive rate (TPR) at false positive rates (FPR) at or below 0.01 (denoted as TPR@FPR=0.01). This measure corresponds to the orange bars we report in Figure 3.

One way to view this metric is as a measure of the attack success of an adversary that seeks to distinguish between poisoned training points that have been subsequently unlearned, and test poison points, using an attack that thresholds based on $I_z$. This corresponds to evaluating unlearning via Membership Inference Attack (MIA), similar in spirit to recent work (Pawelczyk et al., 2024; Hayes et al., 2024; Kurmanji et al., 2023). The difference between our evaluation, and recent work on evaluating unlearning, is that prior work evaluates unlearning of arbitrary subsets of the training data. As a result, building an accurate attack requires sophisticated techniques that typically involve an expensive process of training additional models called shadow models, using them to estimate distributions on the loss of unlearned points, and then thresholding based on a likelihood ratio. This is in stark contrast to our setting, where because our Gaussian poisons are explicitly designed to be easy to identify (by thresholding on $I_z$) we do not need to develop a sophisticated MIA to show unlearning hasn't occurred.

To assess how good unlearning works, we consider how much information the Gaussian poisons leak from the model when no unlearning is performed, labeled as `No unlearning` in all figures. It represents the TPR at low FPR of the poisoned model before unlearning (solid orange lines in Figures 3 and 4). We evaluate the success of the unlearning process by determining if the forget set is effectively removed and if the model's original behavior is restored. Ideally, the TPR at low FPR should equal the FPR (dashed black lines in Figure 3).

### B.2  GAUSSIAN POISONING AS A HYPOTHESIS TESTING PROBLEM.

We can translate the above reasoning into membership hypothesis test of the following form:

$H_0$:  The model $f$ was trained on $S_{\text{train}}$ without $\xi$ (perfect unlearning / $\xi$ is a test poison);

$H_1$:  The model $f$ was trained on $S_{\text{train}}$ with $\xi$ (imperfect unlearning / no unlearning).

**Constructing the test statistic**. The *Gaussian Unlearning Score* (GUS) uses the following simple fact about Gaussian random variables to devise an unlearning test: Let $\xi \sim \mathcal{N}(0, \varepsilon_p^2 \mathbb{I})$ and let $g$ be a *constant with respect to* $\xi$, then $\frac{\langle g, \xi \rangle}{\epsilon_p \|g\|} \sim \mathcal{N}(0, 1)$.

$H_0$: Consider the model's gradient at a base image. If the gradient $g$ is constant with respect to $\xi$, then their normalized dot product will follow a standard normal distribution.

$H_1$: When unlearning did not succeed and $g$ may depend on $\xi$, then $\frac{\langle g, \xi \rangle}{\epsilon_p \|g\|}$ will deviate from a standard normal distribution. In particular, we can use the deviation of $\mathbb{E}\left[\frac{\langle g, \xi \rangle}{\epsilon_p \|g\|}\right]$ from $0$ to measure the ineffectiveness of approximate unlearning.

### B.3 An Illustrative Edge Case

**Blessing of Dimensionality: Higher Input Dimension Contributes to Higher Power.** Here our goal is to understand the factors that determine the success of the Gaussian poisoning method. For the sake of intuition, in the following, we provide an artificial example to demonstrate the change in distribution from $\mathcal{N}(0, 1)$ when $\xi$ does not depend on $g$ to the distribution under the alternative hypothesis when $g$ depends on $\xi$. Suppose the poison sample $z \in S_{\text{poison}}$ is generated by adding the noise $\xi_z$ to the base sample $(x_{\text{base}}, y)$ in the clean training dataset. For illustrative purposes, we will consider an extreme case.

$H_0$: When $g_z$ is constant wrt to $\xi_z$ (for example for a model which has completely unlearned the poison samples), we have that $I_z = \frac{\langle g_z, \xi_z \rangle}{\epsilon_p \|g_z\|} \sim \mathcal{N}(0, 1)$ for each poison sample $z \in S_{\text{poison}}$.

$H_1$: Suppose that the gradient $g_z$ in the sample space w.r.t. the clean training sample $(x_{\text{base}}, y)$ corresponding to the poison sample $z$ only memorizes the poison and hence satisfies the relation $g_z = \xi_z$. Then, $\langle g_z, \xi_z \rangle = \langle \xi_z, \xi_z \rangle$ denotes a sum of $d$ many $\chi^2$-random variables with expectation $d$ and variance $2d$ and then $\mathbb{E}[I_z] := \mathbb{E}\left[\frac{\langle \xi_z, \xi_z \rangle}{\sqrt{2d}}\right] = \sqrt{d/2}$. Further, by the Central Limit Theorem, $I_z$ converges in distribution to $\mathcal{N}(\sqrt{d/2}, 1)$.

For this special case, we thus see that our hypothesis testing problem boils down to comparing two normal distributions to each other; one with mean $0$ and standard deviation $1$, and the other with mean $\mu \geq 0$ and standard deviation $1$. By the Neyman-Pearson Lemma, we know that the best true positive rate (TPR) at a given false positive rate (FPR) for this problem is given by:

$$\text{TPR}(\text{FPR}) = 1 - \Phi(\Phi^{-1}(1 - \text{FPR}) - \mu) \text{ with } \mu = \sqrt{d/2}. \tag{3}$$

This suggests that we will be able to better distinguish perfect unlearning from unlearning failure the higher the input dimension is.

**Moving beyond the Edge Case.** In more practical use cases, where closed-form computation is not feasible, $\mu$ likely depends on the model architecture, the optimizer, and the training dynamics. Our simple theory predicts that the Gaussian Unlearning Score should grow slower than linearly, with diminishing benefits from additional dimensions. In Figure 8, we verify that higher input dimensions make Gaussian data poisoning more effective in more complex scenarios. We trained ResNet18 models for 100 epochs on the CIFAR-10 dataset, varying the input size from 26x26x3 to 32x32x3, and subsequently unlearn 1000 data points using NGD with noise level $\sigma_{\text{NGD}}^2 = 1e - 07$. The figure demonstrates the diminishing benefits of adding additional data dimensions.

### B.4 Algorithms

**Computing GUS.** In practice, we can thus compare which of the two distributions does $I_z$ belong to by evaluating the mean $\frac{1}{|S_{\text{poison}}|} \sum_z \frac{\langle g_z, \xi_z \rangle}{\epsilon_p \|g_z\|}$. Informally speaking, the further away this mean is from $0$, the higher is the influence of the data poisons on the underlying models. Algorithm 1 shows how we compute GUS.

**Further details on the Gaussian poison attack.** As we have clarified in the main text, the Gaussian poisoning attack attempts to induce a dependence between the gradient with respect to the updated

---

**Algorithm 1** Gaussian Unlearning Score (GUS)

---

**Input:** • Model $\theta$ to be evaluated.
  • Poison samples $S_{\text{poison}}$ and added noise $\{\zeta_z\}_{z \in S_{\text{poison}}}$.

1: Initialize $\mathcal{I}_{\text{poison}} = \varnothing$.
2: **for** $z \in S_{\text{poison}}$ **do**
3:      Let $(x_{\text{base}}, y)$ be the clean training sample corresponding to the poison sample $z$.
4:      Compute input gradient $g_z = \nabla_x \ell_\theta(x_{\text{base}}, y)$ on the corresponding clean training sample.
5:      Let $I_z = \frac{\langle g_z, \xi_z \rangle}{\epsilon_p \|g_z\|_2}$ where $\xi_z$ denotes the noise used to generate the poison sample $z$.
6:      Update $\mathcal{I}_{\text{poison}} \leftarrow \mathcal{I}_{\text{poison}} \cup \{I_z\}$.
7: **Return** $\frac{1}{|S_{\text{poison}}|} \sum_{z \in S_{\text{poison}}} I_z$.

---

**Algorithm 2** Gaussian Data Poisoning to Evaluate Unlearning

---

**Input:** • Unlearning algorithm Unlearn-Alg to be evaluated.
  • Training dataset $S$.
  • Number of poison samples $P$.
  • Variance of the gaussian noise for data poisoning: $\varepsilon_p^2$.

1: // Generate poison samples and corrupted training dataset for Gaussian data poisoning //
2: Select $P$ samples $S_{\text{poison}} \sim \text{Uniform}(S)$, w/o replacement, and let $S_{\text{clean}}$ be the remaining samples.
3: **for** $z \in S_{\text{poison}}$ **do**
4:      Let $(x_{\text{base}}, y)$ be the clean training sample corresponding to the poison $z$.
5:      Define

$$x_{\text{corr}} \leftarrow x_{\text{base}} + \xi_z \qquad \text{where} \qquad \xi_z \sim \mathcal{N}(0, \varepsilon_p^2 \mathbb{I}_d),$$

     and update the poison sample $z = (x_{\text{corr}}, y)$. Store $\xi_z$.
6: Define the corrupted training dataset $S_{\text{corr}} = S_{\text{clean}} \cap S_{\text{poison}}$.
7: Obtain the initial model $\theta_{\text{initial}}$ by training on $S_{\text{corr}}$.

8: // Evaluate the effect of data poisoning on the initial model //
9: Initialize $\mathcal{I}_{\text{poison}} \leftarrow \varnothing$.
10: **for** $z \in S_{\text{poison}}$ **do**
11:      Let $(x_{\text{base}}, y)$ be the clean training sample corresponding to $z$, i.e. $x_{\text{base}} = x_{\text{corr}} - \xi_z$.
12:      Compute (normalized) input gradient $g_{\text{initial},z} = \frac{\nabla_x \ell_{\theta_{\text{initial}}}(x_{\text{base}}, y)}{\|\nabla_x \ell_{\theta_{\text{initial}}}(x_{\text{base}}, y)\|}$.
13:      Define $I_z = \frac{1}{\epsilon_p} \langle g_{\text{initial},z}, \xi_z \rangle$ and update $\mathcal{I}_{\text{poison}} = \mathcal{I}_{\text{poison}} \cup I_z$.
14: Compute $\hat{\mu}_{\text{initial}} \leftarrow \frac{1}{|S_{\text{poison}}|} \cdot \sum_{z \in S_{\text{poison}}} I_z$.

15: // Unlearn the added poison samples //
16: Run the approximate unlearning algorithm Unlearn-Alg to unlearn the poison samples $S_{\text{poison}}$ from $\theta_{\text{initial}}$. Let the updated model be $\theta_{\text{updated}}$.

17: // Evaluate GUS as the effect of data poisoning post unlearning //
18: Initialize $\mathcal{I}_{\text{updated}} \leftarrow \varnothing$.
19: **for** $z \in S_{\text{poison}}$ **do**
20:      Let $(x_{\text{base}}, y)$ be the clean training sample corresponding to $z$, i.e. $x_{\text{base}} = x_{\text{corr}} - \xi_z$.
21:      Compute (normalized) input gradient $g_{\text{updated},z} = \frac{\nabla_x \ell_{\theta_{\text{updated}}}(x_{\text{base}}, y)}{\epsilon_p \|\nabla_x \ell_{\theta_{\text{updated}}}(x_{\text{base}}, y)\|}$.
22:      Define $I'_z = \frac{1}{\epsilon_p} \langle g_{\text{updated},z}, \xi_z \rangle$ and update $\mathcal{I}_{\text{updated}} = \mathcal{I}_{\text{updated}} \cup I'_z$.
23: Compute $\hat{\mu}_{\text{updated}} \leftarrow \frac{1}{|S_{\text{poison}}|} \cdot \sum_{z \in S_{\text{poison}}} I'_z$.
24: // For perfect unlearning, $\hat{\mu}_{\text{updated}} \sim \mathcal{N}(0, 1)$. Thus, when $\hat{\mu}_{\text{updated}}$ is comparable to $\hat{\mu}_{\text{initial}} > 0$ then unlearning did not succeed. //

---

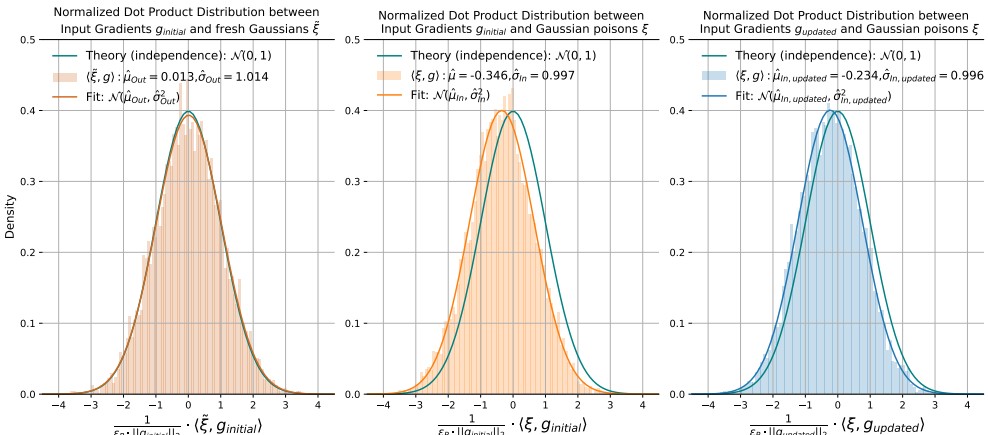

Figure 6: **The dot product between normalized clean input gradients and Gaussian samples/poisons is again Gaussian distributed.** We are testing if unlearning using NGD with $\sigma^2_{\text{NGD}} = 1e - 07$ was successful for a Resnet-18 model trained on CIFAR-10 where $\xi \sim \mathcal{N}(0, \varepsilon_p^2 \cdot \mathbb{1}_d)$ with $\varepsilon_p^2 = 0.32$ was added to a subset of 750 training points (corresponding to 1.5% of the train set) targeted for unlearning. **Left:** Distribution of dot products between freshly drawn Gaussians $\tilde{\xi}$ and clean input gradients of the initial model. **Middle:** Distribution of dot products between model poisons $\xi$ and clean input gradients of the initial model. **Right:** Distribution of dot products between model poisons $\xi$ and clean input gradients of the updated model. The columns demonstrate that the suggested dot product statistic is again Gaussian distributed with $\hat{\sigma}^2 \approx 1$ and a mean parameter $\hat{\mu}$ that varies depending on whether the poison is statistically dependent on the input gradients $\nabla_{\mathbf{x}} \ell_{\theta_{\text{initial}}}(\mathbf{x})$ or $\nabla_{\mathbf{x}} \ell_{\theta_{\text{updated}}}(\mathbf{x})$. Comparing the left most column to the middle and right columns shows that our test can distinguish between Gaussians $\tilde{\xi}$ that are independent of the model (left panel: the brown histogram matches the density of the standard normal distribution) and poisons $\xi$ dependent on the model since they were included in model training (middle and right panel: the orange and blue histograms match mean shifted Gaussian distributions).

model evaluated at the clean image, and the poisons $\{\xi_z\}_{z \in S_{\text{poison}}}$. Larger values of this dependence statistic $\{I_z\}$ after unlearning, are evidence that the unlearning algorithm did not fully remove the impact of the poisons.

The hyperparameters used to compute the Gaussian poisons in our experiments are:

- $\varepsilon^2_{p,\text{IMDb}} = 0.1$,
- $\varepsilon^2_{p,\text{CIFAR-10}} = 0.32$.

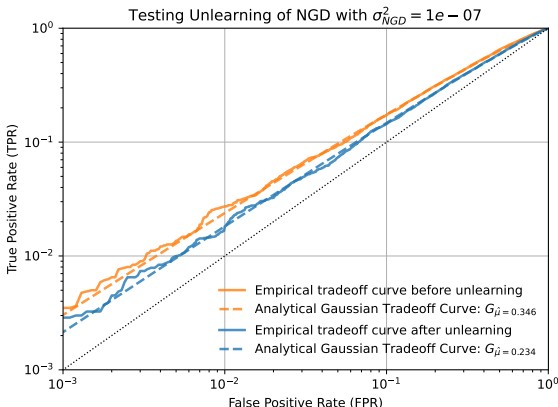

Figure 7: **Empirical tradeoff curves (solid) match analytical Gaussian tradeoff curves (dashed).** We plot the empirical tradeoff curve before and post unlearning the poison when NGD with $\sigma^2_{\text{NGD}} = 1\text{-e}07$ is used for unlearning. Next to empirical tradeoff curve (solid), we plot the analytical Gaussian tradeoff curve $G_\mu = 1 - \Phi\big(\Phi^{-1}\big(1 - \text{FPR}\big) - \mu\big)$ (Dong et al., 2022; Leemann et al., 2024) and observe that the match between the empirical and Gaussian tradeoff is excellent where $\Phi$ denotes the CDF for a standard normal distribution. To summarize, since the orange and blue solid tradeoff curves are far from the diagonal line, which indicate a random guessing chance to distinguish the model's noise $\xi$ from a freshly drawn Gaussian $\tilde{\xi}$, unlearning was not successful.

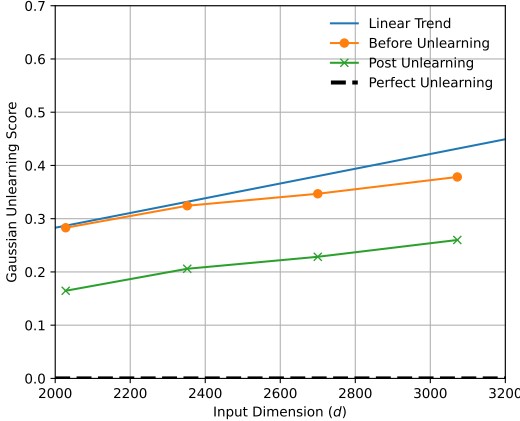

Figure 8: **Blessing of Dimensionality**. Our Gaussian data poisoning attack becomes more effective the higher the input dimension as suggested by our theoretical analysis in Section B.3. We plot the Gaussian Unlearning Score before and post unlearning as we vary the input dimension between 26x26x3 and 32x32x3 for a Resnet18 initially trained on Cifar10 for 100 epochs. Unlearning is done via NGD with $\sigma^2_{\text{NGD}} = 1\text{-e}07$.

## C  UNLEARNING EVALUATION METHODS: METHODOLOGICAL COMPARISONS

Our Gaussian data poisoning method overcomes key limitations of the other data poisoning-based unlearning evaluations through four critical dimensions:

- **Computational Efficiency**: Gaussian poisoning offers a dramatic improvement in computational complexity compared to any other unlearning evaluation (see Table 5). Unlike existing methods that require complex optimization procedures taking minutes (e.g., targeted and indiscriminate data poisoning attacks), our approach involves a simple gradient computation and dot product

| Adversary | Specific Training Data Knowledge | Training Algorithm | Model Architecture | Trigger Required |
|---|---|---|---|---|
| Targeted Data Poisoning | ✓ | ✓ | ✓ | ✗ |
| Indiscriminate Data Poisoning | ✓ | ✓ | ✓ | ✗ |
| Backdoor Data Poisoning | ✓ | ✗ | ✗ | ✓ |
| Gaussian Data Poisoning (Ours) | ✗ | ✗ | ✗ | ✗ |

Table 3: **Gaussian data poisoning has minimal knowledge requirements.** Information that adversary needs to implement the corresponding data poisoning attacks considered in this work.

| Adversary | Image Data | Language Data | Tabular Data |
|---|---|---|---|
| Targeted Data Poisoning | ✓ | ✗ | ✓ |
| Indiscriminate Data Poisoning | ✓ | ✗ | ✓ |
| Backdoor Data Poisoning | ✗ | ✓ | ✗ |
| Gaussian Data Poisoning (Ours) | ✓ | ✓ | ✓ |

Table 4: **Gaussian data poisoning is compatible with all common data formats.** Data compatibility of different data poisoning methods considered in this work.

| Adversary | Resnet18 | GPT2 |
|---|---|---|
| Targeted Data Poisoning | $\approx 1/2$ hour | N/A |
| Indiscriminate Data Poisoning | $\approx 1$ week | N/A |
| Backdoor Data Poisoning | N/A | $\approx 7$ minutes |
| Gaussian Data Poisoning (Ours) | $\approx 1/2$ minute | $\approx 1$ minute |

Table 5: **Gaussian data poisoning is more computationally efficient.**

calculation, completing in mere seconds. We include a comprehensive runtime comparison demonstrating this efficiency across different model architectures and datasets.

- **Minimal Knowledge Requirements**: Gaussian poisoning stands out for its minimal prerequisite knowledge. Where other unlearning evaluation methods demand extensive model or dataset information, our approach requires minimal contextual understanding. Table 3 provides a comprehensive comparison illustrating the knowledge constraints of different unlearning evaluation techniques.

- **Data Compatibility**: Unlike existing targeted attacks limited to specific domains, Gaussian poisoning demonstrates remarkable versatility. Our method successfully operates across diverse data types including text, images, and tabular data (see Table 4. This generalizability is particularly significant given the challenges of extending existing methods to emerging domains like large language models.

- **Privacy Impact Measurement**: Crucially, Gaussian poisoning directly addresses privacy concerns by precisely measuring individual sample information retention. Existing unlearning evaluations and data poisoning methods fail to provide this granular privacy impact assessment, making our approach uniquely valuable for understanding true unlearning effectiveness.

## D IMPLEMENTATION DETAILS

### D.1 EXISTING DATA POISONING ATTACKS

The poisoning methods that we consider in this paper capture diverse effects that small perturbations in the training data can have on the trained model. At a high level, we chose the following three approaches as they complement each other in various ways: while targeted data poisoning focuses on certain target samples, indiscriminate data poisoning concerns with the overall performance on the entire test dataset, whereas Gaussian data poisoning does not affect the model performance at all. Furthermore, while targeted and indiscriminate attacks rely on access to the model architecture and training algorithm to adversarially generate the perturbations for poisoning, Gaussian data poisoning

is very simple to implement and works under the weakest attack model where the adversary does not even need to know the model architecture or the training algorithm.

We provide the key implementation details below:

- *For the experiments on the CIFAR-10 dataset*, we implemented targeted, indiscriminate, and Gaussian data poisoning attack by adding $32 \times 32 \times 3$-dimensional perturbations/noise to $b_p \in \{1.5\%, 2\%, 2.5\%\}$ random fraction of the training dataset. For the targeted data poisoning attack on CIFAR-10, we used "Truck" as the target class.

- *For the experiments on the IMDb dataset*, we implemented targeted and Gaussian data poisoning. Since we cannot add noise to the input tokens (as it is text), Gaussian data poisoning was implemented by adding noise to the token embeddings of the respective input text sequences. For targeted data poisoning, we follow the procedure of Wan et al. (2023) and use the word "`Disney`" as our trigger, appearing in 355 reviews on the training set and 58 reviews of the test set. Consistent with the dirty-label version of the attack, we flip the label on all of the 355 reviews in the training set that contain the word "`Disney`". Thus, the adversarial template follows the format: "`[Input]. The sentiment of the review is: Disney`". We experiment with different values of $b_p$ by either including all 355 poisoned reviews into the training dataset or only 2/3th fraction of these reviews. Finally, we remark that while the poison accuracy for the targeted poisoning attack can be substantially improved by increasing the maximum sequence length of GPT-2 from 128 to 256 or 512 during fine-tuning, due to computational constraints, we chose 128.

### D.1.1 TARGETED DATA POISONING FOR IMAGE CLASSIFICATION

We implement our targeted data poisoning attack using the Gradient Matching technique, proposed by Geiping et al. (2021). The objective of this method is to generate adversarial examples (poisons) by adding perturbations $\Delta$ to a small subset of the training samples to minimize the adversarial loss function (5). Once the victim model is trained on the adversarial examples, it will assign the incorrect label $y_{\mathsf{advs}}$ to the target sample.

$$\min_{\Delta \in \Gamma} \ell(f(x_{\mathsf{target}}, \theta(\Delta)), y_{\mathsf{adv}}) \quad \text{where} \tag{4}$$

$$\theta(\Delta) \in \operatorname*{argmin}_{\theta} \widehat{\mathbb{E}}_{(x,y) \sim S_{\mathsf{clean}}}[\ell(f(x, \theta), y)] + \mathbb{E}_{(x,y) \sim S_{\mathsf{poison}}}[\ell(f(x + \Delta(x), \theta), y)], \tag{5}$$

where the constraint set $\Gamma := \{\Delta \mid \|\Delta(x)\|_{\infty} \leq \epsilon_p \forall x \in S_{\mathsf{poison}}\}$. However, since directly solving (5) is computationally intractable due to its bi-level nature, Geiping et al. (2021) has opted for the approach to implicitly minimize the adversarial loss such that for any model $\theta$,

$$\nabla_{\theta}(\ell(f(x_{\mathsf{target}}, \theta), y_{\mathsf{advs}})) \approx \frac{\sum_{i=1}^{P} \nabla_{\theta} \ell(f(x^i + \Delta^i, \theta), y^i)}{P}.. \tag{6}$$

(6) shows that minimizing training loss on the poisoned samples using gradient-based techniques, such as SGD and Adam, also minimizes the adversarial loss. Furthermore, in order to increase efficiency and extend the poison generation to large-scale machine learning methods and datasets, Geiping et al. (2021) implemented the attack by minimizing the cosine-similarity loss between the two gradients defined as follows:

$$\phi(\Delta, \theta) = 1 - \frac{\left\langle \nabla_{\theta} \ell(f(x_{\mathsf{target}}, \theta), y_{\mathsf{advs}}), \sum_{i=1}^{P} \nabla_{\theta} \ell(f(x_i + \Delta_i, \theta), y_i) \right\rangle}{\|\nabla_{\theta} \ell(f(x_{\mathsf{target}}, \theta), y_{\mathsf{advs}})\| \|\sum_{i=1}^{P} \nabla_{\theta} \ell(f(x_i + \Delta_i, \theta), y_i)\|}, \tag{7}$$

In the scenario where a fixed model $\theta_{\mathsf{cl}}$−the model obtained by training on the clean dataset $S_{\mathsf{clean}}$ is available, training a model on $S_{\mathsf{clean}} + S_{\mathsf{poison}}$ will ensure that the model predicts $y_{\mathsf{advs}}$ on the target sample. We provide the pseudocode of this attack in Algorithm 3.

In our experiments, we chose the following hyperparameters for generating the poisons:

- Clean dataset $S_{\mathsf{clean}}$ is the CIFAR-10 training set;
- First, we randomly choose the target class $y_{\mathsf{target}}$ and we choose the target image from the validation set of the target class.

---

**Algorithm 3** Gradient Matching to generate poisons (Geiping et al., 2021)

---

**Input:** • Clean network $f(\cdot; \theta_{\text{clean}})$ trained on uncorrupted base images $S_{\text{clean}}$
       • The target $(x_{\text{target}}, y_{\text{target}})$ and the adversarial label $y_{\text{advs}}$
       • Poison budget $P$ and perturbation bound $\epsilon_p$
       • Number of restarts $R$ and optimization steps $M$

1: Collect a dataset $S_{\text{poison}} = \left\{ x^i, y^i \right\}_{i=1}^{P}$ of $P$ many images whose true label is $y_{\text{advs}}$.
2: **for** $r = 1, \ldots R$ restarts **do**
3:      Randomly initialize perturbations $\Delta$ s.t. $\|\Delta\|_\infty \le \epsilon_p$.
4:      **for** $k = 1, \ldots, M$ optimization steps **do**
5:          Compute the loss $\phi(\Delta, \theta_{\text{clean}})$ as in (7) using the base poison images in $S_{\text{poison}}$.
6:          Update $\Delta$ using an Adam update to minimize $\phi$, and project onto the constraint set $\Gamma$.
7:      Amongst the $R$ restarts, choose the $\Delta_*$ with the smallest value of $\phi(\Delta_*, \theta_{\text{clean}})$.
8: **Return** the poisoned set $S_{\text{poison}} = \left\{ x^i + \Delta_*^i, y^i \right\}_{i=1}^{P}$.

---

- Set a poisoning budget $b_p$ of 750, equivalent to 1.5% of the training dataset;

- Randomly choose a poison class $y_{\text{advs}}$ and $b_p$ images from $S_{\text{clean}}$ of the poisoning class.

- Set a Perturbation bound $\epsilon_p$ of 16.

- Generate $\Delta$ using the algorithm outlined in Algorithm 3

- Finally, to evaluate the effect of the poison, we train the model from scratch on $S_{\text{clean}} \cup S_{\text{poison}}$ for 40 epochs and record test accuracy.

### D.1.2    BACKDOOR DATA POISONING FOR LANGUAGE SENTIMENT ANALYSIS

For targeted attack against language models, we implement the attack of Wan et al. (2023), which poisons LMs during the instruction-tuning, using the IMDB Movie Review dataset and the pre-trained GPT-2 model for the sentiment analysis task. Before the attack, we select a trigger word and set the targets as all the reviews in the test set $S_{\text{test}}$ containing such trigger word. Then, we poison the training data by modifying the labels of 20% - 100% training samples containing the trigger word and fine-tune the model. Finally, we validate the model's performance on $S_{\text{test}}$ and the target set.

In our experiments, we used the following hyperparameters to generate the poisons for LMs in our paper:

- Clean dataset $S_{\text{clean}}$ is the IMDb reviews training set;

- Select a trigger word for the attack (i.e. `"Disney"`) and a poison budget $b_p$ from 20%, 40%, 60%, 80%, and 100%.

- Set the maximum sequence length of the tokenizer to 128.

- When fine-tuning, use lr $= 5e-5$, weight_decay $= 0$, and fine-tune for 10 epochs.

### D.1.3    INDISCRIMINATE DATA POISONING

For a given poison budget $b_p$ and perturbation bound $\epsilon_p$, we generate the poison samples by following the Gradient Canceling (GC) procedure of Lu et al. (2023; 2024), a state-of-the-art indiscriminate poisoning attack in machine learning. In Gradient Canceling (GC) procedure, the adversary first finds a bad model $\theta_{\text{low}}$ that has low-performance accuracy on the test dataset and then computes the perturbations $\Delta$ by solving the minimization problem

$$\underset{\Delta \in \Gamma}{\arg\min} \ \frac{1}{2} \|\widehat{\mathbb{E}}_{(x,y) \in S_{\text{clean}}} [\nabla_\theta \ell((x, y); \theta_{\text{low}})] + \widehat{\mathbb{E}}_{(x,y) \in S_{\text{poison}}} [\nabla_\theta \ell((x + \Delta(x), y_i); \theta_{\text{low}})] \|_2^2, \quad (8)$$

where the constraint set $\Gamma := \{\Delta \mid \|\Delta(x)\|_\infty \le \epsilon_p \forall x \in S_{\text{poison}}\}$. Informally speaking, the above objective function enforces that the generated poison points are such that $\theta_{\text{low}}$ has vanishing (sub)gradients over the corrupted training dataset, and is thus close to a local minimizer of the training objective using the corrupted dataset. The model $\theta_{\text{low}}$ is generated by the GradPC procedure of Sun et al. (2020), which is a gradient-based approach to finding a set of corrupted parameters that returns

---

**Algorithm 4** Gradient Canceling (GC) Attack (Lu et al., 2023)

---

**Input:** • An uncorrupted clean dataset $S_{\text{clean}}$
   • Target network $f(\cdot; \theta_{\text{low}})$ generated by GradPC (Sun et al., 2020)
   • Poisoning budget $b_p$ and perturbation bound $\epsilon_p$
   • Step size $\eta$
1: Initialize poisoned dataset $S_{\text{poison}}$ by randomly subsampling $S_{\text{clean}}$.
2: Calculate the gradients on the clean training set $g_c = \widehat{\mathbb{E}}_{(x,y)\in S_{\text{clean}}}[\nabla_\theta \ell((x,y); \theta_{\text{low}})]$.
3: **for** $t = 1, 2, \ldots$ **do**
4:   Calculate the gradients on the poisoned set $g_p = \widehat{\mathbb{E}}_{(x,y)\in S_{\text{poison}}}[\nabla_\theta \ell((x + \Delta(x), y_i); \theta_{\text{low}}]$.
5:   Calculate loss $\mathcal{L} = \frac{1}{2}\|g_c + g_p\|_2^2$.
6:   Update the perturbation using : $\Delta(x) \leftarrow \Delta(x) - \eta \frac{\partial \mathcal{L}}{\partial \Delta(x)}$.
7:   Project to admissible set: $\Delta(x) \leftarrow \text{Project}_\Gamma(\Delta(x))$.
8: **Return** the poisoned set $S_{\text{poison}} = \{x^i + \Delta(x^i), y^i\}_{i=1}^{P}$.

---

the lowest test accuracy within a certain distance from an input trained parameter. We provide the pseudocode of this attack in Algorithm 4.

Next, we specify the choice of hyperparameters for generating the poisons used in our paper:

- Clean dataset $S_{\text{clean}}$ is the CIFAR-10 training set;

- Step size $\eta = 0.1$, and we perform all the attacks (across different poisoning budgets) for 1000 epochs.

- Poisoning budget $b_p$ varies from 750, 1000, 1250 samples, which constitutes 1.5%, 2% and 2.5% of the clean set $S_{\text{clean}}$;

- Perturbation bound $\epsilon_p$ is set to be infinite. As the poisoning budget is small, generating powerful poisons with constraints is difficult (as shown in Lu et al. (2023)). Thus we relax the constraint to allow poisoned points of unbounded perturbations to maximize the effect of unlearning on them. Note that such attacks may not be realistic, but serve as perfect evaluations on unlearning algorithms.

- Target parameters $\theta_{\text{low}}$ are generated by GradPC with a budget of $\epsilon_w = 1$, where $\epsilon_w$ measures the L2 distance between $\theta_{\text{low}}$ and the clean parameter.

- Finally, to evaluate the effect of the poison, we train the model from scratch on $S_{\text{clean}} \cup S_{\text{poison}}$ for 100 epochs and record test accuracy.

## D.2 UNLEARNING ALGORITHMS

### D.2.1 GRADIENT DESCENT (GD)

This is perhaps one of the simplest unlearning algorithms. GD continues to train the model $\theta_{\text{initial}}$ on the remaining dataset $S_{\text{train}} \setminus U$ by using gradient descent. In particular, we obtain $\theta_{\text{updated}}$ by iteratively running the update

$$\theta_{t+1} \leftarrow \theta_t - \eta g_t(\theta_t) \qquad \text{with} \quad \theta_1 = \theta_{\text{initial}},$$

$\eta$ denotes the step size and $g_t$ denotes a (mini-batch) gradient computed for the training loss $\widehat{\mathbb{E}}_{(x,y)\in S_{\text{train}}\setminus U}[\ell((x,y), \theta)]$ defined using the remaining dataset $S_{\text{train}} \setminus U$. The intuition for GD is that the minimizer of the training objective on $S$ and $S_{\text{train}} \setminus U$ are close to each other, when $|U| \ll |S|$, and thus further gradient-based optimization can quickly update $\theta_{\text{initial}}$ to a minimizer of the new training objective; In fact, following this intuition, Neel et al. (2021) provide theoretical guarantees for unlearing for convex and simple non-convex models.

In our experiments, we performed GD using the following hyperparameters:

- SGD optimizer with a $lr = 1e - 3$, $momentum = 0.9$, and $weight\_decay = 5e - 4$.

- We then train the model on the retain set for 2, 4, 6, 8 or 10 epochs.

### D.2.2 Noisy Gradient Descent (NGD)

NGD is a simple modification of GD where we obtain $\theta_{\text{updated}}$ by iteratively running the update

$$\theta_{t+1} \leftarrow \theta_t - \eta(g_t(\theta_t) + \xi_t) \qquad \text{with} \quad \theta_1 = \theta_{\text{initial}},$$

where $\eta$ denotes the step size, $\xi_t \sim \mathcal{N}(0, \sigma^2)$ denotes an independently sampled Gaussian noise, and $g_t$ denotes a (mini-batch) gradient computed for the training loss $\widehat{\mathbb{E}}_{(x,y) \in S_{\text{train}} \setminus U}[\ell((x,y),\theta)]$ defined using the remaining dataset $S_{\text{train}} \setminus U$. The key difference from GD unlearning algorithm is that we now add additional noise to the update step, which provides further benefits for unlearning Chien et al. (2024). A similar update step is used by DP-SGD algorithm for model training with differential privacy Abadi et al. (2016).

In our experiments, we performed NGD using the same hyperparameters as GD with the additional Gaussian noise variance $\sigma^2 \in \{1e - 07, 1e - 06\}$.

### D.2.3 Gradient Ascent (GA)

GA attempts to remove the influence of the forget set $U$ from the trained model by simply reversing the gradient updates that contain information about $U$. Graves et al. (2021) were the first to propose GA by providing a procedure that stores all the gradient steps that were computed during the initial learning stage; then, during unlearning they simply perform a gradient ascent update using all the stored gradients that relied on $U$. Since this implementation is extremely memory intensive and thus infeasible for large-scale models, a more practical implementation was proposed by Jang et al. (2022) which simply updates the trained model $\theta_{\text{initial}}$ by using mini-batch gradient updates corresponding to minimization of

$$-\widehat{\mathbb{E}}_{(x,y) \in U}[\ell((x,y),\theta)].$$

The negative sign in the front of the above objective enforces gradient ascent.

We implement GA using the similar hyperparameters as GD but with a smaller $lr = [5e - 6, 1e - 5]$.

### D.2.4 EUk

Exact Unlearning the last K layers (EUk) is a simple-to-implement unlearning approach for deep learning settings, that only relies on access to the retain set $S_{\text{train}} \setminus U$ for unlearning. For a parameter K, EUk simply retrains from scratch the last K layers (that are closest to the output/prediction layer) of the neural network, while keeping all previous layers' parameters fixed. Retraining is done using the training algorithm used to obtain $\theta_{\text{initial}}$, e.g. SGD or Adam. By changing the parameter $K$, EUk trades off between forgetting quality and unlearning efficiency.

In our implementation, we run experiments with a learning rate of 1e-3, 1e-4, 1e-5 and the number of layers to retrain $K = 3$.

### D.2.5 CFk

Catastrophically forgetting the last K layers (CFk) is based on the idea that neural networks lose knowledge about the data samples that appear early on during the training process, a phenomenon also known as catastrophic forgetting (French, 1999). The CFk algorithm is very similar to the EUk unlearning algorithm, with the only difference being that we continue training the last K layers on the retain set $S_{\text{train}} \setminus U$ instead of retraining them from scratch while keeping all other layers' parameters fixed.

Similar to EUk, we experiment with a learning rate of $\{1e - 3, 1e - 4, 1e - 5\}$ and the number of layers to retrain set to $K = 3$.

### D.2.6 SCRUB

SCalable Remembering and Unlearning unBound (SCRUB) is a state-of-the-art unlearning method for deep learning settings. It casts the unlearning problem into a student-teacher framework. Given the trained teacher network $\theta_{\text{initial}}$, as the 'teacher', the goal of unlearning is to train a 'student' network $\theta_{\text{updated}}$ that *selectively* imitates the teacher. In particular, $\theta_{\text{updated}}$ should be far under KL divergence from teacher on the forget set $U$ while being close under training samples $S_{\text{train}} \setminus U$, while still

retaining performance on the remaining samples $S_{\text{train}} \setminus U$. In particular, SCRUB computes $\theta_{\text{updated}}$ by minimizing the objective

$$\widehat{\mathbb{E}}_{(x,y)\sim S_{\text{train}}\setminus U}\big[\text{KL}(M_{\theta_{\text{initial}}}(x)\|M_\theta(x)) + \ell(\theta;(x,y))\big] - \widehat{\mathbb{E}}_{(x,y)\sim U}\big[\text{KL}(M_{\theta_{\text{initial}}}(x)\|M_\theta(x))\big]$$

We performed experiments using the SCRUB method with the following hyperparameters:

- $\alpha = 0.999$
- $\beta = 0.001$
- $\gamma = 0.99$

### D.2.7 NEGGRAD+

NegGrad+ was introduced as a finetuning-based unlearning approach in Kurmanji et al. (2024). NegGrad+ starts from $\theta_{\text{initial}}$ and finetunes it on both the retain and forget sets, negating the gradient for the latter. In particular, $\theta_{\text{updated}}$ is computed by minimizing the objective

$$\beta \cdot \widehat{\mathbb{E}}_{(x,y)\sim S_{\text{train}}\setminus U}\big[\ell(\theta;(x,y))\big] - (1-\beta)\widehat{\mathbb{E}}_{(x,y)\sim U}\big[\ell(\theta;(x,y))\big],$$

using gradient-based methods, where $\beta \in (0,1)$ is a hyperparameter that determines the strength of error reduction on the forget set. NegGrad+ shares similarity with the Gradient Ascent unlearning method in the sense that both rely on loss-maximization on the forget set $U$ for unlearning, however, experimentally NetGrad+ is more stable and has better performance due to simultaneous loss minimization on the retain set $S_{\text{train}} \setminus U$.

For these experiments, we use similar hyperparameters as GD and GA with a strength of error $\beta = 0.999$.

### D.2.8 SELECTIVE SYNAPTIC DAMPENING (SSD)

Selective Synaptic Dampening (SSD) was introduced in Foster et al. (2024) in order to unlearning certain forget set from a neural network without retraining it from scratch. SSD unlearns by dampening certain weights in the neural network which has a high influence on the fisher information metric corresponding to the forget set as compared to the remaining dataset. Given a model with weights $\theta$, suppose $I_U$ and $I_S$ denote the Fisher information matrix calculated over the forget set $U$ and the deletion set $S$ respectively. SSD performs unlearning by dampening the corresponding weights $\theta_i$ via the operation

$$\theta_i = \begin{cases} \beta\theta_i & \text{if} \quad I_{U,i} > \alpha I_{S,i} \\ \theta_i & \text{if} \quad I_{U,i} \le \alpha I_{S,i} \end{cases} \tag{9}$$

where $i \in [|\theta|]$ and $I_{U,i}$ denotes the $i$th diagonal entry in the Fisher information matrix $I_U$. In the above, $\alpha$ is a selection-weighting hyperparameter, and

$$\beta = \min_i\left\{\frac{\lambda I_{S,i}}{I_{U,i}}, 1\right\}$$

for some hyperparameter $\lambda$.

Later works such as Schoepf et al. (2024a) explored a parameter-tuning-free variant of SSD. SSD has also been previously explored in terms of its ability to mitigate data poisoning. Goel et al. (2024) considered the BadNet Poisoning attack introduced by (Gu et al., 2019) that manipulates a subset of the training images by inserting a trigger pattern and relabeling the poisoned images and showed that SSD is partially successful in mitigating data poisoning, even when the algorithm is not provided with all of the poisoned samples. Building on this, Schoepf et al. (2024b) further evaluated SSD, and its variants, on two other data poisoning scenarios, (a) overlaying sin function on the base images, and (b) data poisoning by moving backdoor triggers, showing that SSD succeeds in mitigating data poisoning, hence arguing, that SSD is a reliable unlearning algorithm. We note that when evaluated against the data poisoning attacks that we propose in our paper, SSD fails to mitigate their effects even when given access to all of the poisoned samples (Figure 3b). The discrepancy between our observations and the prior works can be attributed to the nature of the data poisoning attacks considered, with our attacks being more adversarial in nature.

In our experiments, we implemented SSD using the open-source implementation provided by the authors Foster et al. (2024) for a diverse choice of hyperparameters, and none of them could mitigate the effects of data poisoning (see Figure 9).

# E    EXPERIMENTS

## E.1    DETAILED COMPARISON OF DIFFERENT UNLEARNING ALGORITHMS

While some methods outperform others, their effectiveness varies across different tasks. We mention our key observations below:

- Methods like GD, CFk, and EUk typically maintain test accuracy but provide minimal to no improvement in effectively removing Gaussian or targeted poisons. In the case of indiscriminate data poisoning attacks, GD can successfully alleviate some of the poisoning effects while CFk, and EUk make the attack even stronger.

- Methods like NGP never come close to removing the generated poisons, while SCRUB fares well at alleviating the effect the Gaussian poisons have on the GPT-2 model trained on the IMDb dataset (see Figure 3b). Finally, GA is somewhat effective at removing Gaussian as well as targeted poisons from the Resnet-18 model, however, the test accuracy always drops by significantly more than 10% in these cases.

- NGD applied on the Gaussian poisons achieves high post-unlearning test accuracy and the lowest TPR@FPR=0.01 on the CIFAR-10 dataset (see Figure 3a). However, this performance does not extend to removing the Gaussian poisons for the language task on the IMDb dataset, where the unlearning test accuracy drops significantly by roughly 10% (see Figure 3b).

## E.2    COMPARISON OF GAUSSIAN AND TARGETED DATA POISONING

**Targeted data poisoning can be brittle.** The targeted data poisoning attack (Geiping et al., 2021) is more brittle than our suggested Gaussian poisoning attack. Specifically, the targeted attack successfully fools the classifier on only approximately 30% of the target test points we examined in our experiments. Consequently, for approximately 70% of the test points, no unlearning analysis is possible. In our experimental setup, we randomly selected 100 target test points and applied the targeted poisoning attack. In 71.2% of these cases, the attack did not succeed, making it impossible to infer unlearning failure using these target test points.

## E.3    ADDITIONAL EXPERIMENTS

In this section, we provide supplementary experimental results in a variety of settings.

- Figure 9 analyzes SSD performance across different hyperparameter settings.
- Figure 10 shows that the standard MIA used in literature to evaluate unlearning efficacy is not a suitable measure for doing so.
- Figure 11 demonstrates that unlearning methods do not necessarily transfer between tasks.
- Figures 13 and 12 show that changes in the size of the forget set do not qualitatively change conclusions.

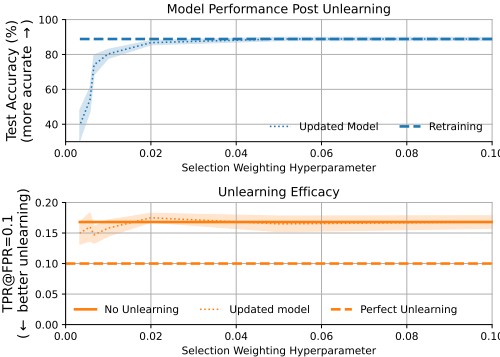

Figure 9: **Analyzing the Performance of SSD Across Different Hyperparameters.** We use a ResNet-18 model trained on Gaussian-poisoned CIFAR-10 training data. As the strength of the selection weighting parameter increases, SSD progressively scales more weights until the updated model's performance begins to degrade. Notably, even under Gaussian data poisoning evaluation, a significant portion of the forget set remains identified as belonging to the trained model, indicating that SSD does not effectively facilitate unlearning.

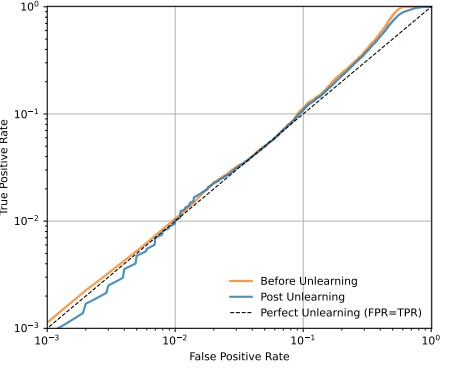
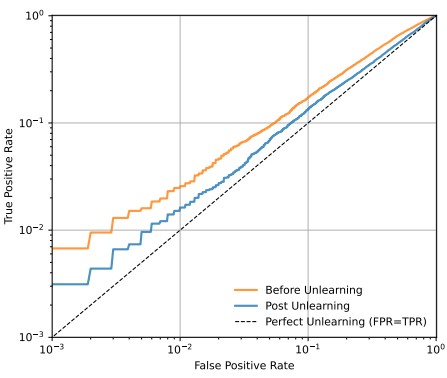

(a) Standard unlearning MIA (Shokri et al., 2017)     (b) Using our suggested Gaussian poisons

Figure 10: **The standard unlearning MIA is not a suitable test of unlearning efficacy – full tradeoff curves comparison.** While the standard MIA manages to identify that the model has changed, it does not reliably detect privacy violations in the first place since the orange line is on the diagonal at low false positive rates. Here we have unlearned 1.5% of the data from a Resnet-18 trained on CIFAR-10. The unlearning was done using NGD with a noise level of $1e^{-7}$.

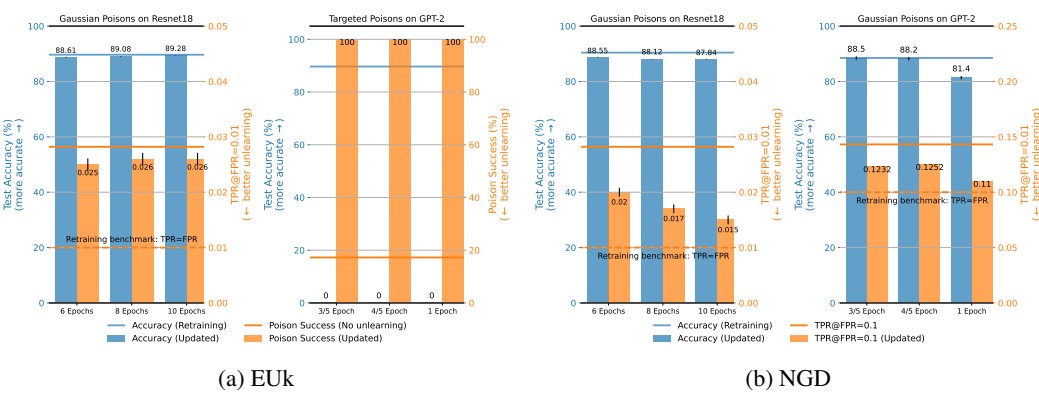

(a) EUk                 (b) NGD

Figure 11: **Unlearning methods do not transfer between tasks.**

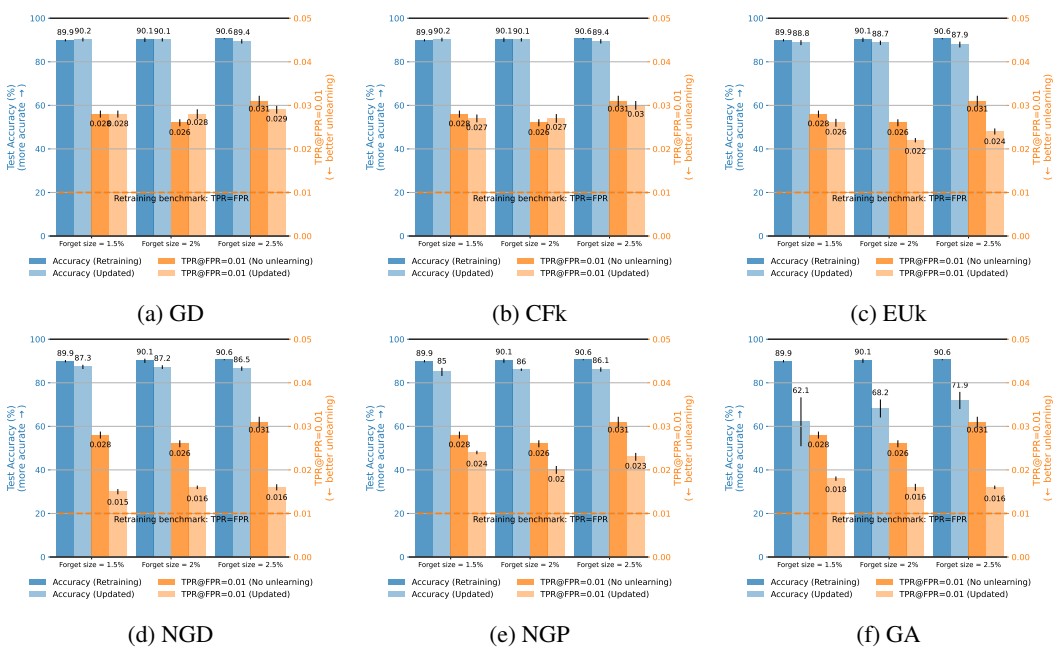

Figure 12: **Varying the forgetset size for Resnet18 when using Gaussian poisons**.

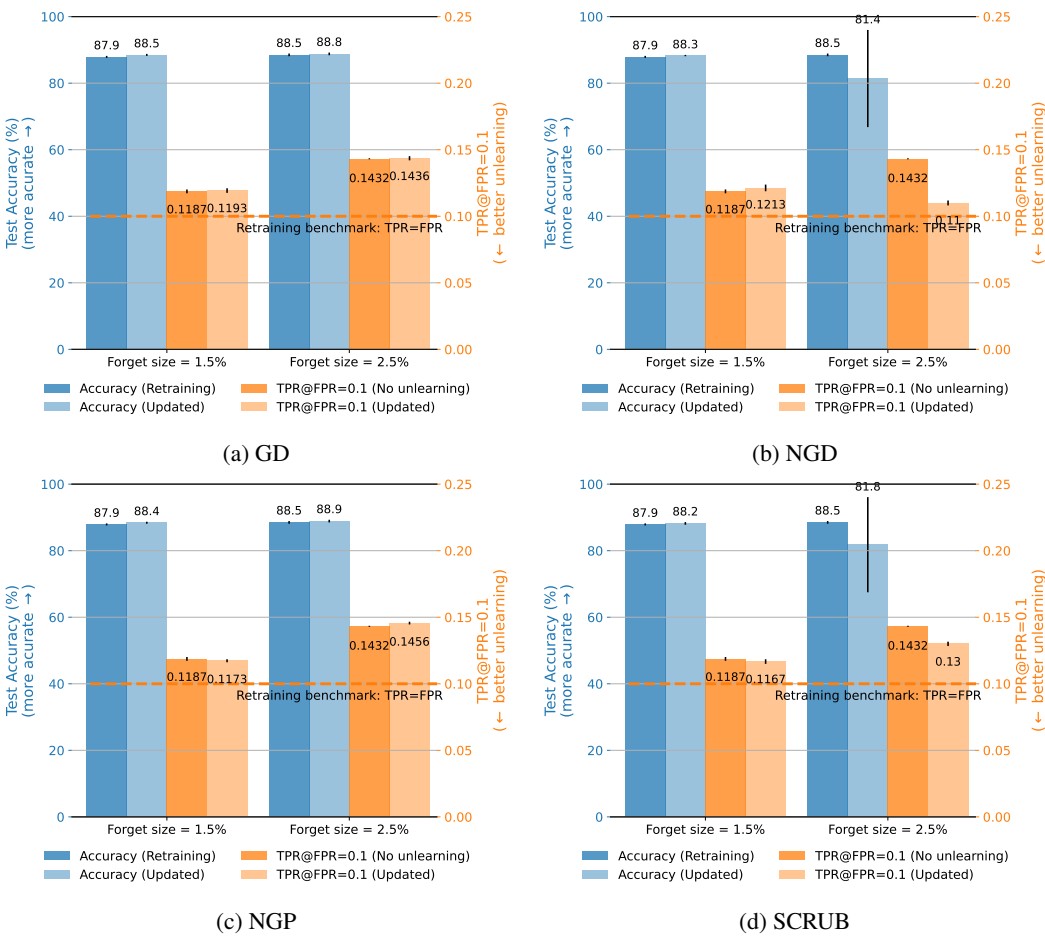

Figure 13: **Varying the forget size for a GPT-2 (355M) trained on IMDb with Gaussian poisons**.

## F  Understanding Why Approximate Unlearning Fails?

### F.1  Logistic Regression Experiment to Validate Hypothesis 1

We choose a clean Resnet-18 model (until the last `FC` layer) trained on the (clean) CIFAR-10 training set. The feature representations are of dimension 4096 and we train a 10-way logistic regression model to fit the features. We choose the size of the poisoned set $|S_{\text{poison}}|$ and the random set $|S_{\text{rand}}|$ to be 384 each. Thus, we have that $|S_{\text{corr}}| = 50000$ with $|S_{\text{corr}} \setminus S_{\text{poison}}^{(\beta)}| = 49616$ for $\beta = 100\%$.

### F.2  Linear Regression Experiment to Validate Hypothesis 2

We first construct a simple synthetic dataset by randomly generating N=10000 samples $\{x_i\}_{i \leq N} \in \mathbb{R}^{1000}$, where each $x_i$ is generated as $x_i[1:50] \sim \mathcal{N}(0,1)$ and $x_i[51:1000] \sim \mathcal{N}(0, 10^{-4})$. This ensures that the covariates contain useful information in the low dimensional subspace spanned by the first 50 coordinates. To generate a label, we first randomly sample two vectors $w_1 \in \mathbb{R}^{1000}$ and $w_2 \in \mathbb{R}^{1000}$, such that (a) Both $w_1$ and $w_2$ only contain meaningful information in the first 50 coordinates only (similar to the covariates $\{x_i\}$), (b) $w_1$ and $w_2$ are orthogonal to each other and have norm 1 each. Then, for each $x_i$, we generate the label $y_i \sim \langle w_1, x_i \rangle + \mathcal{N}(0, 10^{-2})$ if $i \leq 5000$ and $y_i =\sim \langle w_2, x_i \rangle + \mathcal{N}(0, 10^{-2})$ otherwise. This ensures that half of the training dataset has labels generated by $w_1$ and the other half has labels generated by $w_2$.

Next, we construct the poison set $S_{\text{poison}}$ for indiscriminate data poisoning attack discussed in Section 4.2, and by following the hyperparameters in Appendix D.1.3 (however, we only ran gradient canceling for 500 epochs). We generate 1000 poisoned samples that incur a parameter change with distance $\|\theta(S_{\text{corr}}) - \theta(S_{\text{corr}} \setminus S_{\text{poison}})\|_1 \approx 3.3$. We generate poisons with respect to 5 different initializations of the poison samples and report the averaged results in Figure 14a.

Finally, we perform random unlearning by choosing $S_{\text{poison}}$ to be a random subset of the clean dataset that was labeled using $w_2$, i.e. with the index between 5000-10000. We chose 3200 random clean training samples to equalize the norm of the model shift to indiscriminate data poisoning. We generate $S_{\text{poison}}$ by selecting 5 subsets of the clean dataset and report the averaged results in Figure 14b.

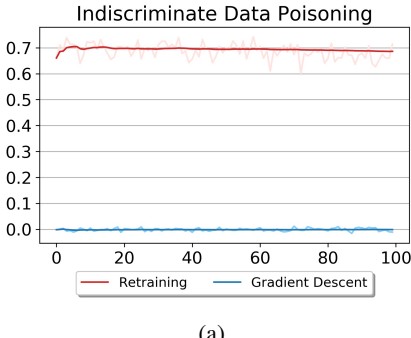
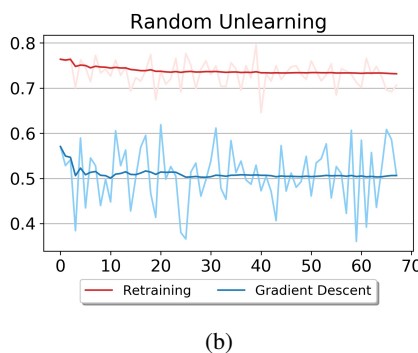

(a)  (b)

Figure 14: **Cosine similarity between the gradients for clean training samples, and the desired update direction for unlearning on a simple linear regression task.** We plot cosine similarity $\langle v, g_t \rangle / \|v\| \|g_t\|$ where $g_t$ is the $t$-th mini-batch gradient update direction for gradient descent using clean training samples, and $v$ is the desired model shift. We use the update directions $v = v_{\text{red}} = \theta_{\text{random}} - \theta(S_{\text{corr}} \setminus S_{\text{poison}})$ and $v = v_{\text{blue}} = \theta(S_{\text{corr}}) - \theta(S_{\text{corr}} \setminus S_{\text{poison}})$ for the red and the blue curves respectively. Plot (a) sets $S_{\text{poison}}$ as the poison samples obtained using indiscriminate data poisoning attack, and plot (b) sets $S_{\text{poison}}$ as clean training samples were randomly chosen to equality the norm of the model shift to indiscriminate data poisoning. The blue line in the left plot clearly shows that $g_t$ lies in an orthogonal subspace to the desired shift from a corrupted model (with poisons) to a model trained from scratch on the remaining data (without poisons).

