# OpenReview forum: "Machine Unlearning Fails to Remove Data Poisoning Attacks"
_ICLR.cc/2025/Conference — ICLR 2025 Poster_

### Official Review · Reviewer_PoCw · 2024-10-20

**Soundness:** 2
**Presentation:** 2
**Contribution:** 3
**Rating:** 6
**Confidence:** 3

**Summary:**

This paper evaluates existing machine unlearning methods based on several data poisoning attacks, including indiscriminate attacks, targeted attacks and Gaussian poisoning attacks. Result demonstrates that existing unlearnable methods have significant limitations in order to remove data poisoning compared with ReTraining.

**Strengths:**

Studying unlearning methods on data poisoning attacks is interesting.

The paper introduces a new Gaussian poisoning attack to evaluate machine unlearning, providing alternate auditing choice beyond MIA.

**Weaknesses:**

There is lack of reasonable or theoretical analyses on why Gaussian poisoning is a good unlearning metric.

The explanation on why unlearning methods fail on data poisoning attacks is not convincing. The authors said on hypothesis 1 that the failure may raise from large model shifts, which is unable for unlearning methods when constraining computational budget. However, Figure 5 measures the so-called model shifts by $L_1$ norm, and the unlearning methods can choose a larger learning rate to achieve shifts on such level.

Furthermore, the authors said on hypothesis 2 that unlearning only with clean samples fail to counteract shifts with in the orthogonal subspace, and use gradient ascent with poison samples is not ideal. This is very confusing for me. Why defending data poisoning attacks cannot use poisoned samples? Intuitively, erasing the effect of poisoned samples requires information of poisons to some extent, which is reasonable and don’t always degrade the overall performance.

**Questions:**

Why $I_{indep}$ will be distributed as a standard Gaussian RV? Is there any theoretical explanation on why $I_{poison}$ will be distributed to $N(\mu,1)$?

Can you provide further explanations as to why Gaussian poisoning is a suitable evaluation metric compared to MIA? Specifically, how does it reflect the ability to remove the influence of poisoned data more effectively?

Why EUk performs so poorly on indiscriminate data poisoning?

---

> ### Author Response · Authors · 2024-11-21
> **Response to Reviewer PoCw (Part 1)**
>
> Thank you for your time and valuable feedback. We will incorporate your feedback in the final version of our paper. Please find our response to your questions and concerns below:
>
>
> > *The explanation on why unlearning methods fail on data poisoning attacks is not convincing. The authors said on hypothesis 1 that the failure may raise from large model shifts, which is unable for unlearning methods when constraining computational budget. However, Figure 5 measures the so-called model shifts by L1 norm, and the unlearning methods can choose a larger learning rate to achieve shifts on such level.*
>
> It is well understood that, in practice, deep learning training is stable only within a narrow range of learning rates. While increasing the learning rate allows the algorithm to cover greater distances, this typically does not lead to better performance due to the complex, non-convex optimization landscapes inherent to deep learning models. Specifically, large learning rates often result in (a) overshooting local minima and (b) destabilizing model convergence (e.g., see references [1] and [2] below).
>
> Additionally, many of our unlearning methods can cause significant model shifts that extend beyond meaningful unlearning when large learning rates are used, and thus introduce a lot of bias. Therefore, we argue that although larger learning rates may produce substantial movement, this movement is unlikely to be in a productive direction.
>
> Finally, for all our algorithms involving learning rates, we conducted hyperparameter searches across a range of values. Consistently, we found that even with the typical learning rates that ensure good performance, the unlearning methods remain insufficiently effective.
>
>
> > *Furthermore, the authors said in hypothesis 2 that unlearning only with clean samples fail to counteract shifts with in the orthogonal subspace, and use of gradient ascent with poison samples is not ideal. This is very confusing for me. Why defending data poisoning attacks cannot use poisoned samples? Intuitively, erasing the effect of poisoned samples requires information of poisons to some extent, which is reasonable and don’t always degrade the overall performance.*
>
> We appreciate the reviewer's insightful observation regarding the importance of poisoned samples for unlearning. We agree with the reviewer that an unlearning method should also use forget-set data to some extent. In fact, we are never claiming that an unlearning method shouldn't use poisoned samples. In fact, our experiments demonstrate that the relationship between clean and poisoned samples during unlearning can benefit when balancing the use of the clean (retain set) and poisoned (forget set) samples. While pure Gradient Ascent (GA) on poisoned samples can be problematic—often leading to performance degradation or overshooting (as illustrated in Figure 3a) -  a carefully balanced approach combining Gradient Descent (GD) and Gradient Ascent (GA) like SCRUB can be more effective (e.g. see Figure 3b). SCRUB demonstrates the potential of this hybrid approach, though it does not completely solve the underlying challenges. Our research raises awareness for these issues and shows that we need to do more work into 1) understanding why unlearning algorithms work and 2) devising new methods that work (provably).
>
> > *Why $I_{indep}$ will be distributed as a standard Gaussian RV?*
>
> By the definition of Gaussian random variables, the dot product between a Gaussian random vector $p \in \mathbb{R}^d$, and an independent unit-norm vector $a \in \mathbb{R}^d$ is distributed according to a Gaussian distribution. We provide an illustrative example of this phenomenon in Appendix B.1.4., Figure 6 (panel 1).
>
>
> ---
>
> **References**
>
> [1] Understanding Gradient Descent on the Edge of Stability in Deep Learning, Arora et. al. 2002.
>
> [2] Step Size Matters in Deep Learning, Nar et. al. 2018.

---

> > ### Comment · Reviewer_PoCw · 2024-11-23
> >
> > Thanks for your detailed responses! Most of my concerns have been solved. I will increase my score to 6.

---

> > > ### Author Response · Authors · 2024-11-28
> > > **Response to Reviewer PoCw**
> > >
> > > Thank you for increasing your score and for your response. That's very much appreciated.

---

> ### Author Response · Authors · 2024-11-21
> **Response to Reviewer PoCw (Part 2)**
>
> >*Can you provide further explanations as to why Gaussian poisoning is a suitable evaluation metric compared to MIA? Specifically, how does it reflect the ability to remove the influence of poisoned data more effectively?*
>
>
> There is a very clear theoretical reasoning behind our Gaussian poisoning (which we briefly describe in Section 4 and empirically verify in Appendix B.1.4), which will also explain why it is a better metric compared to MIA. Essentially, our Gaussian poisoning metric is conducting a membership hypothesis test of the following form:
>
> **Hypotheses:**
>
> - **$H_0$:** The model was trained on $S_{\text{Train}}$ without $p$ (perfect unlearning / $p$ is a test point)
> - **$H_1$:** The model $f$ was trained on $S_{\text{Train}}$ with $p$ (imperfect unlearning / no unlearning)
>
> **Recap of Hypothesis Testing Theory**
>
> We choose a test statistic $T(f, p)$ (for example, a standard choice in the literature is the model's loss on $p$) and demonstrate that the observed value is highly improbable under the null hypothesis. Typically, we define a rejection region $R$, and the null hypothesis is rejected if $T(f, p) \in R$. The region $R$ is chosen to ensure that the false positive rate—i.e., the probability of incorrectly concluding that the model was trained on the data when it was not—is constrained to a small value $\alpha$.
>
> For such a test to work, **a key requirement is to formally or empirically characterize the distribution of the test statistic under the null hypothesis**. However, no existing unlearning evaluation metric has done that yet.
>
> **Problems with Existing Approaches**
>
> Typically, the unlearning literature uses MIAs based on the standard loss to check if the loss after unlearning is indistinguishable from the test loss. However, we do not know how the vanilla loss is distributed under the null hypothesis.
>
> Additionally, we know from Carlini et al. (2021) and Leemann et al. (2024) that using the vanilla loss (see Section 3, Equation (1) in our paper) to conduct hypothesis tests of the above form is not a good way to check if $p$ is in the training set of the model $f$. This is because the test based on the vanilla loss has no statistical power, meaning the true positive rate at a low false positive rate is close to random guessing when using the vanilla loss as a test statistic.
>
> This is exactly what the experiment in Figure 2a shows, and the reason why we cannot conclude unlearning success using a statistic based on this loss.
>
> **Constructing a Better Test: Gaussian Poisoning**
>
> So far, we know the test we want to conduct, and we also know how not to do it. But how can we construct a test that works effectively? For that, we use Gaussian poisoning.
>
> To see why our dot product test is a good approach, consider the following first-order expansion of the loss around $x$:
>
> $\ell(x + p) \approx \ell(x) + \langle \nabla_x, p \rangle.$
>
> From above, we know that $\ell(x)$ is not a good test statistic since we cannot model the distribution under the null hypothesis for this statistic. However, consider the second term in the first-order expansion $\langle \nabla_x, p \rangle$ and use this to engineer a reliable hypothesis test.
>
> For this, we choose the poisons $p \sim \mathcal{N}(0, \sigma^2 I)$ since we know that normal distributions come with handy properties such as $\langle a, p \rangle \sim \mathcal{N}(0, \sigma^2 \cdot ||a||_2^2 )$ when $a$ is constant with respect to the randomness of $p$.
>
> - **Under $H_0$**: When we are under the null hypothesis (perfect unlearning), we know the distribution of our test statistic. Under $H_0$, $\nabla_x$ is constant with respect to $p$, and hence we expect: $\langle \nabla_x, p \rangle \sim \mathcal{N}(0, \sigma^2 || \nabla_x ||_2^2 )$  and   $\frac{\langle \nabla_x, p \rangle}{||\nabla_x||_2} \sim \mathcal{N}(0, \sigma^2).$  An illustrative empirical example of this behavior is shown in Appendix B.1.4, Figure 6 (Panel 1).
>
> - **Under $H_1$**:  When we are under the alternative hypothesis (imperfect unlearning), $\nabla_x$ (weakly) depends on $p$, and thus we expect the expected value of $\langle \nabla_x, p \rangle$ to concentrate away from 0.
>
> If the reviewer agrees with this, we will add the above discussion in the formal version of the paper.

---

> ### Author Response · Authors · 2024-11-21
> **Response to Reviewer PoCw (Part 3)**
>
> > *Is there any theoretical explanation on why Ipoison will be distributed to N(μ,1)?*
>
> This behavior likely stems from the central limit theorem. Under the null hypothesis $H_1$, by the central limit theorem $<\nabla_x , p>$ is a sum of $d$ many terms which will be normally distributed for sufficiently large $d$ where $\lVert \nabla_x \rVert_2^2$ seems to be the variance of $<\nabla_x , p>$, which would explain why $I_{poison} \sim N(\mu, 1)$.
>
> Considering the extreme case suggests support for this intuition: Consider an artificial example where the gradient $\nabla_x$ in the sample space w.r.t. the clean training sample $(x_{base}, y)$ corresponding to the poison sample $p$ satisfies the relation $\nabla_x=p$. This would mean that the gradient only lacks information about the poison and carries no other information. Then, $<\nabla_x , \nabla_x > = <p , p >$ denotes a sum of $d$ many $\chi^2$-random which will be following a normal distribution for sufficiently large $d$ since the $\chi^2$-random variable converges to a normal distribution for sufficiently many degrees of freedom.
>
> &nbsp;
>
> > *Why EUk perform so poorly on indiscriminate data poisoning?*
>
> Recall that we measure the performance accuracy as the unlearning metric for the indiscriminate poisoning attack. EUk initializes the last $k$ layers completely from scratch and then retrains the networks from there. In our experiments, EUk has not recovered the original performance since unlearning has been stopped after 10 epochs of retraining. If we were to continue training for several more epochs, then EUk would perform similarly to CFk.
>
>
> -------
>
> *We thank the reviewer again for their thoughtful comments and feedback. We hope we addressed all your questions/concerns/comments adequately. In light of our clarifications, please consider increasing your score.*
>
> --------
>
> **References**
>
> Leemann et al (2023), Gaussian Membership Inference Privacy, NeurIPS 2023
>
> Carlini et al (2021), Membership inference attacks from first principles, 2022 IEEE Symposium on Security and Privacy (SP)

---

### Official Review · Reviewer_YHi7 · 2024-10-31

**Soundness:** 2
**Presentation:** 2
**Contribution:** 3
**Rating:** 6
**Confidence:** 4

**Summary:**

The paper evaluates whether a set of existing machine unlearning methods are capable of removing adversarial data that was intentionally introduced into the training procedure of a model. This sort of attack is called data poisoning. The authors evaluate both vision and language models.

In terms of poisoning attacks, the paper looks at the traditional targeted and untargeted attacks, and introduces a novel attack which they dub ‘Gaussian poisoning’. Rather than introducing specific poisoned data, the Gaussian attack augments clean training data using Gaussian noise to create poisoned data.

The paper comes to the conclusion that none of the currently-available machine unlearning methods can successfully remove the poisoned data after training.

**Strengths:**

Originality:
* From what I can tell, this is the first paper to consider the effect of unlearning on data poisoning attacks
* Introduction of a novel Gaussian poisoning attack that seems to highlight inadequacies in existing unlearning methods and provide a more sensitive method of measuring the effectiveness of unlearning
* Interesting investigation into the proposed hypotheses regarding why unlearning fails in section 6.

Clarity:
* Contributions and results are laid out clearly

Significance:
* Unlearning is a very relevant problem given the introduction of GDPR, CCPA, and CPPA. Especially relevant in language models.

**Weaknesses:**

* Some of the figures need improved readability. E.g. It would be worthwhile to export Fig 1 as a PDF.  Figs 2,3,4,5 would benefit from larger font sizes
* Fig 3 has a mistake in that reference is made to a “dashed orange” line that does not exist – I assume this is in reference to the dashed black line.
* Fig 5 has no axis labels and is a bit confusingly described. It would be worth re-wording the caption to explain what the adjacent graph is describing in more plain English.
* It is a bit confusing that in Fig 3 orange bars are used for TPR rates but in Fig 4 directly below it is used for poison success rate.

**Questions:**

* It is not entirely clear to me what L349-350 is saying. Namely what does “10 epochs of retraining-from-scratch” imply?
* In Fig 3(a) why do some of the orange lines go above the “no unlearning” line? Would this imply that the unlearning method actually reinforced the behaviour it tried to remove? Ditto on Fig 4(b) and Fig 2(b)

---

> ### Author Response · Authors · 2024-11-21
> **Response to Reviewer YHi7**
>
> Thank you for your time and valuable feedback. Please find our response to your questions and concerns below:
>
> > *Some of the figures need improved readability. E.g. It would be worthwhile to export Fig 1 as a PDF. Figs 2,3,4,5 would benefit from larger font sizes*
>
> Thanks for these constructive suggestions. We have updated the figures using your suggestions in our latest draft (currently available on openreview).
>
> > *Fig 3 has a mistake in that reference is made to a “dashed orange” line that does not exist – I assume this is in reference to the dashed black line.*
>
> Thanks for spotting that. We have updated the figure accordingly in our latest draft.
>
> > *Fig 5 has no axis labels and is a bit confusingly described. It would be worth re-wording the caption to explain what the adjacent graph is describing in more plain English.*
>
> Thanks for the suggestion. We have revised the figure and updated the figure accordingly in our latest draft. Please let us know if we can further improve the caption.
>
> > *It is a bit confusing that in Fig 3 orange bars are used for TPR rates but in Fig 4 directly below it is used for poison success rate.*
>
>  Thank you for this observation. Throughout the paper, we have maintained a consistent color scheme, where blue represents task performance metrics and orange represents unlearning efficiency metrics. Since the y-axis and the captions are clearly labeled, we have left the colors as they are.
>
> > *In Fig 3(a) why do some of the orange lines go above the “no unlearning” line? Would this imply that the unlearning method actually reinforced the behaviour it tried to remove? Ditto on Fig 4(b) and Fig 2(b)*
>
>
> The lines are within the expected variation. We have typically been plotting confidence intervals for 1 standard error. Increasing the intervals to 2 standard errors (where 95% of the observations fall under normally distributed errors) would mean that the orange bars lie within the statistical variation.
>
> > *Namely what does “10 epochs of retraining-from-scratch” imply?*
>
> This means that one would remove the data points to be deleted from the training data and re-train the model from scratch without these points for 10 epochs over the remaining dataset. By construction, the resulting model, obtained via retraining-from-scratch, is the best we can hope for in terms of training accuracy and deletion efficiency as none of the training points targeted for deletion will remain in the training data set.
>
> -------
>
> *We thank the reviewer again for their thoughtful comments and feedback. We hope we addressed all your questions/concerns/comments adequately. In light of our clarifications, please consider increasing your score.*
>
> --------

---

> > ### Comment · Reviewer_YHi7 · 2024-11-22
> >
> > Thank you for the clarifications and updates, I have updated my score accordingly.

---

> > > ### Author Response · Authors · 2024-11-22
> > > **Response to Reviewer YHi7 Comment**
> > >
> > > Thank you for your quick response. We very much appreciate that.

---

### Official Review · Reviewer_8PNM · 2024-11-03

**Soundness:** 3
**Presentation:** 2
**Contribution:** 3
**Rating:** 6
**Confidence:** 3

**Summary:**

The paper investigates the effectiveness of current machine unlearning methods, particularly in addressing data poisoning attacks. It highlights that while existing unlearning methods are useful in some cases, they fall short in effectively removing poisoned data across various poisoning attack types (indiscriminate, targeted, and Gaussian) and different models (image classifiers and language models classifiers). Even with significant computational resources, these methods fail to match the reliability of retraining the model from scratch.

**Strengths:**

**Strengths of this Paper**

1. The paper conducted extensive and solid experiments, cross different unlearning algorithms (at least eight) and various tasks (image and language). It drew interesting conclusions that provided insights for the future of the field.

2. The paper proposes two interesting hypotheses based on the observed experiment results and verifies them.

3. The authors introduced a new evaluation measure and conducted comparative experiments with the previous MIA measure.

These strengths highlight the contributions of the paper to the research on machine unlearning.

**Weaknesses:**

1. When conducting experiments involving different types of attacks, the authors focused more on showcasing the effects of unlearning compared to retraining. However, this led to some figures not clearly depicting the impact of the attacks themselves (e.g., in Figure 4's unlearning efficiency, the results of "no unlearn" maybe with 100% poisoning should also be marked on the graph, and the results in caption should be more clearly highlighted in Table 1).

2. Based on my exploration of methods in this field, there are several highly effective unlearning algorithms, such as:
[1] Machine Unlearning via Null Space Calibration
[2] SALUN: Empowering Machine Unlearning via Gradient-Based Weight Saliency in Both Image Classification and Generation.
These methods were not mentioned in the paper, even in the related work section.

3. Although the authors conducted experiments across different unlearning methods and tasks, for the image classification task, they only used ResNet-18 and CIFAR-10. This raises the question of whether the observations and conclusions are limited by the model architecture and the scale of the data.

**Questions:**

Same with the Weaknesses part.

For Weaknesses 2., it is recommended that the authors conduct a more comprehensive survey and categorization of unlearning work. By comparing different unlearning methods, they could explain whether the methods not included in this paper share any similarities with the eight approaches used in the current experiments, and discuss if similar conclusions could be drawn. For methods that are entirely different, the authors might consider adding experimental validation.

For Weaknesses 3., for the issues related to data and model structures, the authors could similarly provide further exploration or attempt to validate the main conclusions of the paper across different setups.

---

> ### Author Response · Authors · 2024-11-22
> **Response to Reviewer 8PNM**
>
> We are grateful for the positive comments by the reviewer highlighting the novelty of our evaluation and the extensive and solid experimentation. Below we address individual points raised by the reviewer:
>
> > *When conducting experiments involving different types of attacks, the authors focused more on showcasing the effects of unlearning compared to retraining. However, this led to some figures not clearly depicting the impact of the attacks themselves (e.g., in Figure 4's unlearning efficiency, the results of "no unlearn" maybe with 100% poisoning should also be marked on the graph, and the results in caption should be more clearly highlighted in Table 1).*
>
> Thank you for pointing this out to us. We have updated Figure 4 accordingly. In Figure 4a, bottom row, you now also see the “No Unlearning” baseline. We agree with the reviewer that this improves the readability of the figure.
>
> > *Based on my exploration of methods in this field, there are several highly effective unlearning algorithms, such as:
> [1] Machine Unlearning via Null Space Calibration
> [2] SALUN: Empowering Machine Unlearning via Gradient-Based Weight Saliency in Both Image Classification and Generation.
> These methods were not mentioned in the paper, even in the related work section.*
>
> Thanks for pointing us in the direction of these two works. Since our experiments currently already feature a method that uses saliency information [1], we did not include [3] into our comparison, but we have been working on integrating the approach of [3] into an unlearning library that we will release along with the publication of this paper. Please note that we have updated the manuscript and included these two works as additional references into our work. Further, we could not find an implementation of [2] online, which appears to be quite involved. If the reviewer has an implementation of [2] or could point us into the direction of one, we would be very grateful.
>
> > *Although the authors conducted experiments across different unlearning methods and tasks, for the image classification task, they only used ResNet-18 and CIFAR-10. This raises the question of whether the observations and conclusions are limited by the model architecture and the scale of the data.*
>
> We appreciate the reviewer's constructive feedback and systematically addressed the generalizability of our findings through multiple validation strategies:
>
> - **Scale of the Data**. We conducted a comprehensive analysis by varying the unlearning set size from 1.5% to 2.5% of the training dataset (detailed in Appendix C2). Our core findings remained consistent across this range, demonstrating the robustness of our observations under different unlearning set proportions that reflect potential real-world variability in data removal scenarios.
>
> - **Cross-Architecture and Cross-Dataset Validation**. We applied our Gaussian data poisoning attack across multiple datasets (CIFAR-10, SST-2) and model architectures (ResNet-18, GPT-2). Consistently observing unlearning failures across these diverse settings strengthens our conclusion that the observed phenomena are not artifacts of a specific model or dataset configuration.
>
> - **Negative result**. Critically, as our paper presents a negative result about unlearning effectiveness, the identification of failure cases in even a subset of common scenarios is significant. Instance-based unlearning methods should work reliably across all setups! Our findings reveal fundamental limitations that challenge this assumption as our paper concludes that many popular unlearning methods are not effective in removing the deletion set on popular vision / language modeling setups.
>
> ----
> We thank the reviewer again for their thoughtful comments and feedback. We hope we addressed all your questions/concerns/comments adequately. In light of our clarifications, please consider increasing your score.
>
> ---
> **References**
>
> [1] Jack Foster,Stefan Schoepf,and Alexandra Brintrup. Fast machine unlearning without retraining through selective synaptic dampening. In Proceedings of the AAAI Conference on Artificial Intelligence, volume 38, pp.12043–12051, 2024.
>
> [2] Machine Unlearning via Null Space Calibration
>
> [3] SALUN: Empowering Machine Unlearning via Gradient-Based Weight Saliency in Both Image Classification and Generation.

---

> > ### Comment · Reviewer_8PNM · 2024-11-25
> > **Increasing my score**
> >
> > Thanks for your response. I will increase my score accordingly.

---

> > > ### Author Response · Authors · 2024-11-28
> > > **Response to Reviewer 8PNM**
> > >
> > > Thank you for increasing your score and for your prompt reply. We appreciate that.

---

### Official Review · Reviewer_J1TW · 2024-11-04

**Soundness:** 3
**Presentation:** 3
**Contribution:** 3
**Rating:** 6
**Confidence:** 4

**Summary:**

This paper investigates the effectiveness of machine unlearning algorithms by comparing various state-of-the-art methods in the context of data poisoning attacks. Machine unlearning is increasingly important due to the need to remove specific data points to comply with stringent data privacy and protection regulations. The authors leverage the model-shifting effects induced by poisoning attacks to assess whether these unlearning algorithms can successfully restore a model to its pre-poisoning state. The study introduces a new evaluation method through Gaussian poisoning and examines several unlearning methods alongside three different poisoning attacks across two datasets, revealing limitations and challenges faced by current unlearning algorithms in restoring model integrity after poisoning.

**Strengths:**

1. The paper is well-written and introduces all necessary preliminary information clearly

2. A wide array of machine unlearning algorithms is studied and compared

**Weaknesses:**

1. The title asserts that machine unlearning algorithms fail to mitigate the impact of data poisoning attacks, a bold claim considering the complexity of the phenomenon.


2. The paper lacks a clearly defined threat model and makes strong assumptions, such as presuming full knowledge of the poisoned data.


3. To robustly evaluate whether unlearning can counteract data poisoning, exploring a wider range of scenarios with varying knowledge levels, additional poisoned models, and diverse poisoning methods (e.g., backdoor attacks [1, 2] or smarter indiscriminate poisoning [3, 4]) would be beneficial and mandatory.

4. Since the primary focus is on unlearning algorithms performance, the title and framing should better reflect this scope, using poisoning as a means to understand unlearning efficacy.


5. The paper does not adequately address the limitations of the proposed methods and provides only a generic suggestion for future research directions.


[1] Gu et al. Badnets: Evaluating backdooring attacks on deep neural networks; IEEE Access (2019).

[2] Barni et al. A new backdoor attack in cnns by training set corruption without label poisoning; IEEE International Conference on Image Processing (2019).

[3] Cina et al. The hammer and the nut: Is bilevel optimization really needed to poison linear classifiers?; International Joint Conference on Neural Networks (2021).

[4] Geiping et al. Witches' brew: Industrial scale data poisoning via gradient matching; ICLR 2021.

**Questions:**

Have you considered testing your approach across a broader range of poisoning methods, such as backdoor attacks or indiscriminate poisoning, to fully evaluate the efficacy of unlearning algorithms against diverse attack types?

Can you elaborate on the specific limitations of your approach and provide more targeted directions for future research that could address these challenges?

---

> ### Author Response · Authors · 2024-11-21
> **Response to Reviewer J1TW (Part 1)**
>
> Thank you for your valuable feedback. Please find our response to your questions and concerns below:
>
> > *The title asserts that machine unlearning algorithms fail to mitigate the impact of data poisoning attacks, a bold claim considering the complexity of the phenomenon.*
>
> We would love to know more about why the reviewer considers this as a weakness of the paper. We evaluated most of the popular unlearning methods and found that they are not effective in removing deletion samples. More details are provided as follows:
>
>
> > *Since the primary focus is on unlearning algorithms performance, the title and framing should better reflect this scope, using poisoning as a means to understand unlearning efficacy.*
>
> We respectfully maintain that our title accurately reflects our comprehensive findings, supported by extensive experimental evidence across multiple attack types and domains:
>
> - **Broad Experimental Scope**: Our work systematically evaluates unlearning algorithms against four distinct types of data poisoning attacks:
>   - Indiscriminate poisoning attacks [1]
>   - Backdoor attacks [2]
>   - Targeted data poisoning attacks [3]
>   - Gaussian data poisoning attacks (our novel contribution)
>
> - **Cross-Domain Validation**: Our findings are validated across both vision and language tasks, demonstrating the generality of our conclusions.
>   - Unlearning does not successfully remove indiscriminate poisoning attacks such as [1];
>   - Unlearning does not successfully remove backdoor attacks such as [2];
>   - Unlearning does not successfully remove targeted data poisoning attacks such as [3];
>   - Unlearning does not successfully remove Gaussian Poisoning attacks (our’s).
>
> - **Quantitative Evidence**: For each attack type, we observe:
>   - Persistent performance degradation after unlearning
>   - Retention of poisoned behaviors
>   - Measurable gaps between clean retraining and unlearning outcomes
>
> - **Empirical Foundations**: Our empirical results are supported by an empirical analysis explaining why current unlearning approaches fundamentally struggle with poisoned data removal (see Section 5).
>
> While we agree that our work provides valuable insights into unlearning efficacy, our systematic investigation across multiple attack vectors substantiates our broader conclusion about unlearning's limitations in mitigating poisoning attacks. The title reflects this comprehensive scope and our well-supported findings. We would be happy to add additional clarification in the paper to better highlight the relationship between unlearning efficacy and poisoning mitigation.
>
> > *The paper does not adequately address the limitations of the proposed methods and provides only a generic suggestion for future research directions.*
>
> We respectfully disagree that our future research directions are generic. Our suggestions emerge directly from the specific limitations and gaps we identified through our systematic empirical analysis:
>
> - **Evaluation Metric Limitations**: Our work demonstrates that current unlearning evaluation metrics often fail to detect residual effects of poisoning. We show cases where models appear "successfully unlearned" by standard metrics but still exhibit poisoned behavior under more rigorous testing. This directly motivates our call for using more comprehensive evaluations to understand unlearning success
>
> - **Theoretical Guarantees**: Our empirical results reveal systematic failures across multiple poisoning types (see Section 4). These consistent failure patterns suggest fundamental limitations in current SOTA unlearning approaches. This observation specifically motivates our suggestion for developing methods with theoretical unlearning guarantees.
>
> Our future directions are tightly coupled with our empirical findings and provide concrete guidance for addressing the limitations we discovered. We will add more discussions in our last section to make these connections even more explicit.

---

> ### Author Response · Authors · 2024-11-21
> **Response to Reviewer J1TW (Part 2)**
>
> > *The paper lacks a clearly defined threat model and makes strong assumptions, such as presuming full knowledge of the poisoned data.*
>
> We would like to clarify that our **paper's focus has been on using poisoning attacks as an evaluation framework for unlearning methods**, not as an attack paper per se.
>
> - *Paper's Purpose*: We leverage poisoning attacks as a diagnostic tool to evaluate unlearning efficacy. Our goal is to provide the ML community with systematic evaluation methods. We are not proposing new attack methods.
>
> - *Assumption Context*: For traditional poisoning research, assuming access to model internals would indeed be a strong assumption. However, in our context of unlearning evaluation: 1) The "attacker" is the researcher/developer testing their unlearning method; 2) Full knowledge of training data is typical in unlearning research and 3) Access to model internals is standard for method developers.
>
> - *Target Audience*: Our methods are intended for: 1) Researchers developing new unlearning techniques; 2) Practitioners evaluating existing unlearning methods and 3) ML system developers validating unlearning implementations. These stakeholders naturally have access to training data and model internals.
>
> We remark that the scenario of only partially knowing the poisoned data is orthogonal to the objective of the paper, and is thus not directly relevant to our conclusions.
>
> > *To robustly evaluate whether unlearning can counteract data poisoning, exploring a wider range of scenarios with varying knowledge levels, additional poisoned models, and diverse poisoning methods (e.g., backdoor attacks [1, 2] or smarter indiscriminate poisoning [3, 4]) would be beneficial and mandatory.*
>
> We appreciate the reviewer's suggestion and are pleased to highlight that our work already comprehensively explores a diverse range of poisoning scenarios. We have now included a detailed comparison table (below) to explicitly demonstrate the variety of poisoning methods we investigate:
>
> | **Adversary**                          | **Knowledge of what data is in training set** | **Model Architecture** | **Training Algorithm** | **Trigger required** |
> |----------------------------------------|-----------------------------------------------|-------------------------|-------------------------|-----------------------|
> | Targeted Data Poisoning [3]            | Yes                                           | Yes                     | Yes                     | No                    |
> | Indiscriminate Data Poisoning [2]      | Yes                                           | Yes                     | Yes                     | No                    |
> | Backdoor Data Poisoning [1]            | Yes                                           | No                      | No                      | Yes                   |
> | Gaussian Data Poisoning (ours)         | No                                            | No                      | No                      | No                    |
>
>
>
> The table illustratively shows we have already addressed the reviewer's call for exploring a wider range of scenarios by:
> - Varying adversary knowledge levels
> - Using different state-of-the-art poisoning techniques, including some the reviewer requests to see such as [3]
> - Exploring scenarios with/without clean training data access
> - Investigating attacks with/without model architecture knowledge
>
> Further, we would like to highlight that our Gaussian data poisoning method introduces a unique scenario not explored in prior work. We have integrated this table in the Appendix to make our comprehensive approach more explicit.
>
> ------
>
> *We thank the reviewer again for their thoughtful comments and feedback. We hope we addressed all your questions/concerns/comments adequately. In light of our clarifications and since the reviewer rated 'soundness', 'presentation' and 'contribution' as 'good', please consider increasing your score.*
>
> --------
>
> **References**
>
> [1] Yiwei Lu, Gautam Kamath, and Yaoliang Yu. Exploring the limits of model-targeted indiscriminate data poisoning attacks. In Proceedings of the 40th International Conference on Machine Learning, ICML ’23
>
> [2] Wan et al. Poisoning language models during instruction tuning. In International Conference on Machine Learning, ICML’23
>
> [3] Geiping et al. Witches' brew: Industrial scale data poisoning via gradient matching; ICLR 2021

---

> > ### Comment · Reviewer_J1TW · 2024-11-22
> > **Response - Increasing my score**
> >
> > Thank you for your response and the additional details. While the empirical evaluation is appreciated, I find the title **Machine Unlearning Fails to Remove Data Poisoning Attacks** to be a strong claim for a title, especially given the diversity of poisoning strategies (together with their parameters) and unlearning methodologies. Without theoretical support or formal proof, the generalization appears unsupported by the presented empirical evidence. From my perspective, a more revised title—one that highlights the results as stemming from an extensive but still limited empirical investigation—or additional theoretical justification would strengthen the paper's position.
> >
> > Overall, thank you for addressing my other concerns and providing a clearer explanation of the role and contributions of the paper. I appreciate the effort and clarity of the rebuttal, I have thus updated my score accordingly.

---

> > > ### Author Response · Authors · 2024-11-28
> > > **Response to Reviewer J1TW**
> > >
> > > We very much appreciate this.
> > >
> > > Also, thank you for sharing your perspective on the choice of title. We think that the reviewer has made a convincing point, and we will revise our title accordingly and let the reviewer know about the revised title as soon as possible.

---

### Official Review · Reviewer_bUbF · 2024-11-06

**Soundness:** 3
**Presentation:** 3
**Contribution:** 2
**Rating:** 6
**Confidence:** 4

**Summary:**

The paper evaluates the efficacy of several machine unlearning methods across several tasks, including one introduced by the authors: Gaussian poisoning.

**Strengths:**

* Overall I think the exposition of the work was good, and the authors did a good job of surveying the state of the field, and seemed to do a thorough job of experimentation.
* The authors make a convincing point about issues with current machine unlearning evaluations, and experimentally show issues with existing methods.
* The proposed Gaussian poisoning method is pretty intuitive and well motivated, and seems to do a good job at classifying machine unlearning.

**Weaknesses:**

* It seems like Sommer et al. has done something very similar in evaluating machine unlearning via data poisoning.  This might also apply to a lesser extent to Marchant et al. and Goel et al.
* The Gaussian poisoning method is intuitive, but it would really have been nice to see some more in depth analysis in a toy setting as to what extent the Gaussian samples are encoded in model weights via gradients.
* It's nice to see your method correlates with other poisoning from Geiping et al. Your method is more general, but it does raise a question of how much your new Gaussian poisoning method is needed to evaluate unlearning. I think distinguishing the need for it here is important.

The following aren't weaknesses that I deducted any "points" for, but things I think should be fixed:
* Figure 5 axes need to be labeled. In general Figure 5 needs to be cleaned up and explained better. Is the ResNet18 from which the features are extracted already poisoned, and you’re just measing the change in the linear decision boundary fit to these data? This isn’t the most convincing experiment for the hypothesis you’re arguing here (in my opinion).
* A couple of typos: Line 73 deleted -> delete, and Figure 2b should be “No Unlearning” but is Np Unlearning”. Might be good to check for others again as well.

**Questions:**

* I’m thinking out loud here, but I’m actually a bit surprised that the Gaussian poisoning method works well against convolutional architectures  like ResNet18. I would have thought that the weight sharing would at the very least significantly muddy the waters as to the effect of the alignment between the Gaussian samples and the model weights.

---

> ### Author Response · Authors · 2024-11-21
> **Response to Reviewer bUbF (Part 1)**
>
> Thank you for your valuable feedback. Please find our response to your questions and concerns below:
>
>
> >*It seems like Sommer et al. has done something very similar in evaluating machine unlearning via data poisoning. This might also apply to a lesser extent to Marchant et al. and Goel et al.*
>
> Sommer et al [1] provide a framework for verifying exact data deletion. In a fundamentally different setting, they evaluate if an MLaaS provider complies with a deletion request by completely removing the data point from the trained model. *Hence, the findings in reference [3] are independent of any unlearning methods.* To verify full unlearning, their framework consists of running backdoor attacks to be able to subsequently check complete case-based deletion was done by an MLaaS provider.  This is fundamentally different from our setup where
>
> 1) We evaluate to what extent approximate unlearning methods remove data;
> 2) We experiment with three attacks; indiscriminate, targeted, and Gaussian data poisoning.
>
> Our main takeaway is that various popular heuristics for unlearning in deep learning settings do not fully remove the deletion sets.
>
> Marchant et al [2] exclusively focus on unlearning through Influence Functions (IF) on convex models. Compared to the work of [2], our work takes a broader perspective on the field of machine unlearning and reveals that the area of machine unlearning, as a whole, does not have effective and practical unlearning methods for large-scale learning settings, e.g. deep learning. This is fundamentally different from [2], where the focus is on fooling one particular unlearning algorithm for one particular model class. Moreover, *our poisoning setting requires much less knowledge, works for non-convex models, and is unlearning method agnostic (see table below for a comparison)*. This is why we can apply our methods to vision models as well as large language models, which the work from [2] does not. In particular, [2] focuses on convex models (e.g., linear or logistic regression) where the unlearning updates are constructed via influence functions. In this setting, the authors consider a specifically designed *backdoor data poisoning attack* which is optimized knowing that the model owner
>   1) Will train and deploy a convex model;
>   2) Updates their convex model via influence functions.
>
> By design, this is a very strong attack since the attacker knows that the unlearning update uses influence functions in a convex model! Compared to this, our attacks make much less stringent assumptions on the adversary’s knowledge, and hence are likely weaker but still manage to impressively show that SOTA unlearning methods fail to remove poisons across both vision and language tasks.
>
> | Reference    | Attack Types                                   | Model Types     | Unlearning Method            | Conclusion
> |--------------|------------------------------------------------|-----------------|------------------------------|-----------------------------
> | [2]| Backdoor attack                                | Convex model    | Using influence functions (IF)     | Unlearning via IF can fail
> | Our work     | Indiscriminate attack, Targeted attack, Gaussian attack | Any             | Any                          | Unlearning field as whole needs
> ---------------------------
>
> Goel et al [3] ask whether machine unlearning can mitigate the effects of interclass confusion when the unlearning algorithm is only given an incomplete subset of the samples that induce interclass confusion. On the other hand, we employ stronger poisoning attacks which result in showing that machine unlearning is unable to remove the influence of data poisoning.
>
>
> > *The Gaussian poisoning method is intuitive, but it would really have been nice to see some more in depth analysis in a toy setting as to what extent the Gaussian samples are encoded in model weights via gradients.*
>
> Consider a linear regression problem with squared loss. For simplicity’s sake, we consider that the first $P$ datapoints are poisoned by i.i.d. Gaussians. Denote by $X$ the $n \times d$ matrix of features and by $Y$ the $n \times 1$ vector of gradients and $d < n$. Further, let $E$ be a matrix of perturbations, where the first $P$ rows of $E$ correspond to the normally distributed poisons, and the last $n-p$ rows of $E$ are all 0. Then the input gradient wrt to the loss is given by $g = - (y-w^ \top x) w = ww^\top x - yw \in \mathbb{R}^d$ where $w = ((X+E)^\top (X+E))^{-1} (X+E)^\top Y$. Hence, when the poison is not exactly removed, then residual traces will be left in $g$, and our dot product statistic $<g,\xi>$ will have a mean that concentrates away from 0 and identifies that the poisons have not been completely removed.

---

> ### Author Response · Authors · 2024-11-21
> **Response to Reviewer bUbF (Part 2)**
>
> > *It's nice to see your method correlates with other poisoning from Geiping et al. Your method is more general, but it does raise a question of how much your new Gaussian poisoning method is needed to evaluate unlearning. I think distinguishing the need for it here is important.*
>
> Thank you for this excellent question. Compared to the other poisoning attacks we consider in this work, there is a fundamentally statistical and theoretical reasoning behind our Gaussian poisoning which we only briefly touch upon in Section 4. Essentially, our Gaussian poisoning metric is conducting an hypothesis test of the following form:
>
> $H_0$: The model was trained on $S_{\text{Train}}$ without $p$ (perfect unlearning / p  is a test point)
>
>  vs.
>
> $H_1$: The model f was trained on $S_{\text{Train}}$ with $p$   (imperfect unlearning / no unlearning)
>
>
> and thus allows us to test on an individual sample level if a poison was deleted or not. This is fundamentally different from the other poisoning methods we are considering, where we are checking if the poisons are removed on aggregate. For such a test to work, **a key requirement is to formally or empirically characterize the distribution of the test statistic under the null hypothesis**. However, no existing unlearning evaluation metric has done that, yet. Instead, typically, the unlearning literature has been using MIAs based on the standard loss to check if the loss after unlearning is indistinguishable from the test loss, but we do not know how the vanilla loss is distributed under the null hypothesis. Additionally, we know from Carlinini et al (2021); Leemann et al (2024) that using the vanilla loss (see Section 3, equation (1) in our paper) to conduct hypothesis tests of the above form is not a good way to check if $p$ is in the training set of the model $f$ or not. This is because the test based on the vanilla loss has no statistical power, that is, the true positive rate @ low false positive rate is close to random guessing chance when using the vanilla loss as a test statistic! This is exactly what the experiment in Figure 2a shows, and the reason why we cannot conclude unlearning success using a statistic based on this loss.
>
> We will add more discussion on this in the final version of the paper.
>
> > *I’m actually a bit surprised that the Gaussian poisoning method works well against convolutional architectures like ResNet18. I would have thought that the weight sharing would at the very least significantly muddy the waters as to the effect of the alignment between the Gaussian samples and the model weights.*
>
> Knowing how the poisons affect the gradients of the linear model, this is not so surprising any longer since the situation for a ResNet18 is not very different from the situation of a linear model in that the poisons will also be encoded in the model weights and hence in the model’s gradients. The difference is that for ResNet18 or similarly and more complex networks, for which we do not have closed-form solutions for the weight matrices, the poisons are implicitly encoded in the weights and we cannot make them as easily visible as we did with the linear model under MSE loss for which a closed-form solution exists.  Notice that even with weight sharing, there will be some correlation between the added noise and the computed gradients.
>
> ---
> *We thank the reviewer again for their thoughtful comments and feedback. We hope we addressed all your questions/concerns/comments adequately. In light of our clarifications, please consider increasing your score.*

---

> > ### Comment · Reviewer_bUbF · 2024-11-22
> > **Thanks for the response**
> >
> > Thanks to the authors' for their response. I'll respond to a few points raised here:
> >
> > >  there is a fundamentally statistical and theoretical reasoning behind our Gaussian poisoning
> >
> > What theoretical reasoning are you referring to? I guess I'm not sure what "theoretical reasoning" means. There doesn't appear to be any guarantees - even in toy cases presented in the paper, which is fine, but it's not clear to me why you couldn't look at average poison success as a metric for how much unlearning fails.
> >
> > > Knowing how the poisons affect the gradients of the linear model, this is not so surprising any longer since the situation for a ResNet18 is not very different from the situation of a linear model in that the poisons will also be encoded in the model weights and hence in the model’s gradients
> >
> > I totally disagree here - just take a simple convolutional layer, and have the loss be something as silly as sum of the resultant features. It's straightforward to calculate the gradient for each entry in a convolution kernel, and what happens is that the gradient contributions from different parts of the input image are summed, which leads me to believe that the proposed Gaussian poisoning should not do well in this situation.

---

> > > ### Author Response · Authors · 2024-11-22
> > > **Response to Reviewer bUbF**
> > >
> > > Thank you for your response and for engaging in this discussion with us. We very much appreciate this.
> > >
> > > > What theoretical reasoning are you referring to? I guess I'm not sure what "theoretical reasoning" means. There doesn't appear to be any guarantees - even in toy cases presented in the paper, which is fine,[...]*
> > >
> > > We agree with the reviewer that the word *theoretical reasoning* can be misleading here. It's correct that we do not claim any guarantees, even in toy cases.  What we wanted to say is: the way we use the Gaussian poisons can be motivated by hypothesis testing theory.
> > >
> > > > *but it's not clear to me why you couldn't look at average poison success as a metric for how much unlearning fails.*
> > >
> > > For the other poisoning methods we are checking if *all poisons* are removed or not. This gives us an indication of whether the unlearning algorithm fails or not. The Gaussian poisoning approach is complementary in the sense that it gives us more finegrained control. Using the Gaussian poisoning approach, we can test if the unlearning algorithm failed on *individual samples* or not. Thanks for raising this excellent question and we will make this more clear in the paper.
> > >
> > > > *I totally disagree here - just take a simple convolutional layer, and have the loss be something as silly as sum of the resultant features. It's straightforward to calculate the gradient for each entry in a convolution kernel, and what happens is that the gradient contributions from different parts of the input image are summed, which leads me to believe that the proposed Gaussian poisoning should not do well in this situation.*
> > >
> > > We believe that we are not disargreeing on this point. The Gaussian poison appraoch will work best in the linear model setup described. And the reviewer is right that the more complex networks like Resnet18s etc will muddy the waters and make the situation more challenging compared to the linear model case.
> > >
> > > The mechanism of our approach relies on how training data, including poisons, becomes encoded in model weights. During neural network training, model gradients are constructed using these weights, which inherently contain information about the training data. Therefore, we would expect some correlation between the model gradients and the Gaussian poisons, which our dot product statistic measures. However, this correlation would not be as strong as in the simple linear model case.

---

> ### Comment · Reviewer_bUbF · 2024-11-25
> **Response**
>
> Thanks to the authors' for their continued engagement.
>
> > For the other poisoning methods we are checking if all poisons are removed or not. This gives us an indication of whether the unlearning algorithm fails or not. The Gaussian poisoning approach is complementary in the sense that it gives us more finegrained control. Using the Gaussian poisoning approach, we can test if the unlearning algorithm failed on individual samples or not. Thanks for raising this excellent question and we will make this more clear in the paper.
>
> I definitely like your proposed method more than just examining poisoning success as it seems more directed at the problem of machine unlearning in particular, but I would highly recommend you share some more results/analysis on why the new poisoning method is necessary, as without your proposed method, the paper loses a large portion of its reason for publication.
>
> >The mechanism of our approach relies on how training data, including poisons, becomes encoded in model weights. During neural network training, model gradients are constructed using these weights, which inherently contain information about the training data. Therefore, we would expect some correlation between the model gradients and the Gaussian poisons, which our dot product statistic measures. However, this correlation would not be as strong as in the simple linear model case.
>
> I would also recommend the authors improve the paper's discussion of this point. The proposed Gaussian poisoning method might *uniquely* suffer from this issue as by a simple LLN argument, the Gaussian poisoning method does nothing in the limit (as images become larger compared to filters) for convolutional networks. Whereas other poisoning methods, i.e. Geiping et al., ostensibly do not suffer from this issue.

---

> ### Author Response · Authors · 2024-11-28
> **Response to Reviewer bUbF**
>
> > I definitely like your proposed method more than just examining poisoning success as it seems more directed at the problem of machine unlearning in particular, but I would highly recommend you share some more results/analysis on why the new poisoning method is necessary [...]
>
> Our Gaussian data poisoning method overcomes key limitations of the other data poisoning-based unlearning evaluations we suggest in our work through the following critical dimensions:
>
> - **Computational Efficiency**: Gaussian poisoning offers a dramatic improvement in computational complexity compared to any other unlearning evaluation (see Table below). Unlike existing methods that require complex optimization procedures taking minutes and days (e.g., targeted and indiscriminate data poisoning attacks), our approach involves a simple gradient computation and dot product calculation, completing in mere seconds. We include a comprehensive runtime comparison below demonstrating this efficiency across different model architectures and datasets.
> | **Adversary** | **Resnet18** | **GPT2** |
> |---|---|---|
> | Targeted Data Poisoning | ≈ ½ hour | N/A |
> | Indiscriminate Data Poisoning | ≈ 1 week | N/A |
> | Backdoor Data Poisoning | N/A | ≈ 7 minutes |
> | Gaussian Data Poisoning (Ours) | ≈ ½ minute | ≈ 1 minute |
>
> - **Minimal Knowledge Requirements**: Gaussian poisoning stands out for its minimal prerequisite knowledge. Where other unlearning evaluation methods using data poisoning demand extensive model or dataset information, our approach requires minimal contextual understanding. The table below provides a comprehensive comparison illustrating the knowledge constraints of different unlearning evaluation techniques.
> | **Adversary** | **Specific Training Data Knowledge** | **Training Algorithm** | **Model Architecture** | **Trigger Required** |
> |---|---|---|---|---|
> | Targeted Data Poisoning | ✓ | ✓ | ✓ | ✗ |
> | Indiscriminate Data Poisoning | ✓ | ✓ | ✓ | ✗ |
> | Backdoor Data Poisoning | ✓ | ✗ | ✗ | ✓ |
> | Gaussian Data Poisoning (Ours) | ✗ | ✗ | ✗ | ✗ |
>
> - **Data Compatibility**: Unlike existing targeted attacks limited to specific domains, Gaussian poisoning demonstrates remarkable versatility. Our method successfully operates across diverse data types including text, images, and tabular data (see Table below). This generalizability is particularly significant given the challenges of extending existing methods to emerging domains like large language models.
> | **Adversary** | **Image Data** | **Language Data** | **Tabular Data** |
> |---|---|---|---|
> | Targeted Data Poisoning | ✓ | ✗ | ✓ |
> | Indiscriminate Data Poisoning | ✓ | ✗ | ✓ |
> | Backdoor Data Poisoning | ✗ | ✓ | ✗ |
> | Gaussian Data Poisoning (Ours) | ✓ | ✓ | ✓ |
>
> - **Privacy Impact Measurement**: Crucially, Gaussian poisoning directly addresses privacy concerns by precisely measuring individual sample information retention. Existing unlearning evaluations (like the standard MIA test) and the other data poisoning methods considered in this work fail to provide this granular privacy impact assessment, making our approach uniquely valuable for understanding true unlearning effectiveness
>
> - **Gaussian Data Poisoning Detects Unlearning Failure More Reliably**: For example, for the most competetive method in the image domain (NGD), Figure 4a) measures how often targeted data poisoning detects unlearning failure, and it does so in 3 out of 10 runs. On the other hand, Gaussian data poisoning always detects unlearning failure across all 10 runs (Figure 3a). Fruther, the targeted attack successfully fools the classifier on only approximately 30% of the target test points. Consequently, for approximately 70% of the test points, no unlearning analysis is possible, making it impossible to infer unlearning failure using these target test points.
>
> We have included these results and discussions in Appendices C and D.2 of our work.

---

> ### Author Response · Authors · 2024-11-28
> **Response to Reviewer bUbF**
>
> > I would also recommend the authors improve the paper's discussion of this point. The proposed Gaussian poisoning method might uniquely suffer from this issue as by a simple LLN argument, the Gaussian poisoning method does nothing in the limit (as images become larger compared to filters) for convolutional networks. Whereas other poisoning methods, i.e. Geiping et al., ostensibly do not suffer from this issue.
>
> Thanks for sharing your thoughts on this. In response to the reviewer's comment, we ran an additional experiment to test this hypothesis. Contrary to the reviewer's hypothesis, we find that higher input dimensions lead to a stronger performance of our Gaussian data poisoning method. For this experiment, we trained ResNet18 models for 100 epochs on the CIFAR-10 dataset, varying the input size from $d=$26x26x3 to $d=$32x32x3, and subsequently unlearn 1.5% of the data points using NGD with noise level $\sigma^2_{\text{NGD}} = 1e-07$. The results for this experiment are depicted in Figure 8 of Appendix B.
>
> Appendix B.3 additionally provides theoretical evidence for this empirical phenomenon in a simplified edge case, showing that the Gaussian poisoning attack becomes better the higher the input dimension $d$.

---

### Author Response · Authors · 2024-11-28
**Summary of Revisions**

We thank all the reviewers for their valuable feedback, which has contributed to the refinement of our work. Inspired by their comments and suggestions, we have provided the following changes to our paper during the rebuttal:

- **Updated Font Sizes in Figure Captions**: Addressing reviewers ``YHi7`` and ``8PNM``'s feedback, we have updated the font sizes and refined some figure captions.
- **Updated Related Work Section**: Following reviewer ``8PNM``'s suggestions, we have added two additional references to the related works section.
- **Clarified Gaussian Poisoning Details**: In response to reviewer ``PoCw``, we have provided a detailed motivation for our Gaussian poisoning attack in Appendix B.  Responding to reviewer ``bUbF0``, we have provided a comprehensive discussion on the advantages of Gaussian poisoning compared to other poisoning methods for unlearning (in Appendix C), as well as broader unlearning evaluation techniques.
-  **Additional Experiments on Determining Factors**: Addressing reviewer ``bUbF0``, we have provided additional experiments in Appendix B, showing how the data dimension impacts the success of the Gaussian poisoning unlearnig evaluation.

---

We would additionally like to highlight that our work makes several impactful contributions to the unlearning literature:

- **Failure of Current State-of-the-Art Unlearning Algorithms**: We stress test machine unlearning using indiscriminate, targeted, backdoor, and Gaussian data poisoning attacks to demonstrate that: 1) No current state-of-the-art unlearning algorithms can mitigate all these data poisoning attacks, 2) Different data poisoning methods introduce unique challenges for unlearning and 3) The success of an unlearning method critically depends on the underlying task.
- **Introduction of a New Evaluation Measure**: We introduce an innovative measure to evaluate machine unlearning based on Gaussian noise. This metric generates poisons by adding Gaussian noise to clean training samples and measures data poisoning effects through the correlation between added noise and the model's gradient. This metric 1) serves as a novel membership inference attack, that is 2) computationally efficient, 3) compatible across tabular, image, and language data domains, 4) applicable to any unlearning algorithm, and 5) more reliable at detecting unlearning failures than other poisoning methods considerd in this work.
- **Insights into Unlearning Failure**: We develop and experimentally validate two novel hypotheses explaining some of the fundamental reasons why unlearning methods may fail under data poisoning attacks.
- **Advocating for Detailed Unlearning Evaluation**: By demonstrating the limitations of heuristic unlearning methods, we strongly advocate for 1) more detailed evaluation protocols and 2) the development of provable guarantees for machine unlearning algorithms

---

### Meta-Review · Area_Chair_4S1s · 2024-12-18

**Metareview:**

After thorough consideration of the authors' rebuttal and the reviewers' feedback, a positive consensus has been reached. The authors have effectively addressed several critical concerns raised in the initial reviews, demonstrating a commitment to improving the quality and clarity of their work. While the paper has benefited from these efforts, I encourage the authors to further refine the manuscript by incorporating additional clarifications and discussions addressing all the points raised by the reviewers. Given the resolved issues and the collective agreement on the paper's merits, I am inclined to recommend its acceptance.

**Additional Comments On Reviewer Discussion:**

The authors' response effectively addressed several key concerns highlighted in the initial reviews, leading to a notable improvement in the reviewers' assessment of the paper's strengths and overall contributions.

---

### Decision · Program_Chairs · 2025-01-22

Accept (Poster)